# A Probabilistic Hard Concept Bottleneck for Steerable Generative Models

**María Martínez-García**[*]
Dept. of Computer Science
Saarland University

**Ricardo Vazquez Alvarez**
Dept. of Signal Theory and Communications
Universidad Carlos III de Madrid

**Alejandro Lancho**
Dept. of Signal Theory and Communications
Universidad Carlos III de Madrid

**Pablo M. Olmos**
Dept. of Signal Theory and Communications
Universidad Carlos III de Madrid

**Isabel Valera**
Dept. of Computer Science
Saarland University

## Abstract

Concept Bottleneck Generative Models (CBGMs) incorporate a human-interpretable concept bottleneck layer, which makes them interpretable and steerable. However, designing such a layer for generative models poses the same challenges as for concept bottleneck models in a supervised context, if not greater ones. Deterministic mappings from the model inner representations to soft concepts in existing CBGMs: (i) limit steerable generation to modifying concepts in existing inputs; and, more importantly, (ii) are susceptible to *concept leakage*, which hinders their steerability. To address these limitations, we first introduce the Variational Hard Concept Bottleneck (VHCB) layer. The VHCB maps probabilistic estimates of binary latent variables to hard concepts, which have been shown to mitigate leakage. Remarkably, its probabilistic formulation enables direct generation from a specified set of concepts. Second, we propose a systematic evaluation framework for assessing the steerability of CBGMs across various tasks (e.g., activating and deactivating concepts), which allows us to empirically demonstrate that the VHCB layer consistently improves steerability.

## 1 Introduction

In recent years, a new line of research has emerged to improve the interpretability of Generative Models (GMs) by leveraging Concept Bottleneck Models (CBMs) (Koh et al., 2020). A CBM is an inherently interpretable model consisting of a concept predictor, which maps inputs to human-understandable concepts, and a label predictor, which maps these concepts to task outputs. CBMs offer two key advantages: (i) predictions are explicitly grounded in concepts, and (ii) users can intervene on concepts to steer the output. However, CBMs face two key challenges that limit steerability: concept incompleteness, when the concept set fails to capture all task-relevant information, and concept leakage, when soft concept probabilities unintentionally encode task information. Incompleteness can be addressed with a *side channel* that encodes task information not explained by the concepts (Havasi et al., 2022; Espinosa Zarlenga et al., 2022; Sawada & Nakamura, 2022; Yuksekgonul et al., 2023), while leakage can be reduced by using *hard* concept representations (Margeloiu et al., 2021; Havasi et al., 2022; Lockhart et al., 2022; Vandenhirtz et al., 2024; Sun et al., 2024).

Concept Bottleneck Generative Models (CBGMs) extend CBMs from classification to generative tasks (Ismail et al., 2024; Kulkarni et al., 2025). In this framework, a Concept Bottleneck (CB) layer is introduced at an intermediate location of the generative model, mapping the inner representation of the generator to a set of human-understandable concepts. These methods aim to make generative

---

[*]`martinez-garcia@cs.uni-saarland.de`

models inherently interpretable and, most importantly, steerable. By manipulating concepts in the latent space it is possible to control generation, either by (i) generating new data according to specific concept configurations (*steerable generation*), or (ii) intervening concepts in existing observations (*steerable conditional generation*). Current CBGM methods rely on soft concepts and deterministic CB layers, remaining vulnerable to concept leakage and restricting steerability to concept interventions, as they do not model a generative process over the concept space.

**Contributions.** We address these limitations by introducing the Variational Hard Concept Bottleneck (VHCB) layer, defining the first probabilistic CBGM with hard concepts. The VHCB is based on a binary Variational Autoencoder (VAE), producing probabilistic estimates of binary latent variables that are directly mapped to hard concepts (Section 3). Moreover, this probabilistic formulation enables direct generation from specified concept configurations while still supporting concept inference and steerability through concept interventions. In addition, we introduce a systematic evaluation framework for CBGMs across various inference and steerable generation tasks (Section 4). This comprehensive evaluation allows us to analyze accuracy in concept prediction, alignment between intended and actual outcomes of concept manipulation, distributional shifts on non-target concepts during intervention, and the impact of correlations and biases present in the training data in the behavior of the model (Section 5).

## 2 CBGM FRAMEWORK

In CBGMs, a CB module is inserted at an intermediate location of a generative model to map inner representations to a set of human-interpretable concepts (Ismail et al., 2024; Kulkarni et al., 2025). In general, CBGMs consist of three components, as shown in Figure 1. The pre-bottleneck module ($\boldsymbol{u} \to \boldsymbol{w}$) maps a (noisy) latent input $\boldsymbol{u}$ (usually Gaussian distributed) to a pre-bottleneck embedding $\boldsymbol{w}$. The concept bottleneck module ($\boldsymbol{w} \to (\boldsymbol{c}, \boldsymbol{s}) \to \hat{\boldsymbol{w}}$) maps $\boldsymbol{w}$ to a set of predefined concepts $\boldsymbol{c}$ and an unsupervised embedding $\boldsymbol{s}$. The embedding $\boldsymbol{s}$ acts as a *side channel* to capture information not represented by the concepts, addressing the inherent incompleteness of the concept set in generative tasks, since generation depends on factors that cannot be fully described by human-interpretable concepts (e.g., textures, lighting, or object position). The CB then maps $(\boldsymbol{c}, \boldsymbol{s})$ to a post-concept embedding $\hat{\boldsymbol{w}}$. Finally, the post-bottleneck module ($\hat{\boldsymbol{w}} \to \boldsymbol{x}$) generates the final output $\boldsymbol{x}$. In this setting, one can adopt either an in-hoc approach, training the base generative model (pre- and post-bottleneck modules) together with the CB layer, or a post-hoc approach, where the CB layer is inserted into a pretrained generative model.

**Concept label sources.** We consider a set of $K$ human-interpretable binary concepts $\boldsymbol{y} \in \{0,1\}^K$ indicating the presence of specific features in observations $\boldsymbol{x}$. For example, in the image shown in Figure 1, 'smiling' and 'black hair' would be active, while 'has mustache' and 'wearing sunglasses' would be inactive. Concept labels can be obtained from three sources: (i) annotated datasets $(\boldsymbol{x}_i, \boldsymbol{y}_i)_{i=1}^N$, (ii) pretrained supervised classifiers that predict $\boldsymbol{y}$ from $\boldsymbol{x}$, or (iii) pretrained zero-shot classifiers that infer concepts without task-specific training (Kulkarni et al., 2025).

**Desiderata**. The goal is to design generative models that produce high-quality outputs while remaining both interpretable and steerable. *Generation quality* requires the model to produce diverse data that accurately reflects the training distribution. *Interpretability* requires accurate inference of human-understandable concepts. *Steerability* requires precise control over the generative process: in steerable generation, the model should produce samples reflecting specific concept configurations; in conditional steerable generation, it should be possible to modify target concepts in existing observations $\boldsymbol{x}$ while keeping remaining concepts unchanged. Additionally, disentanglement between the unsupervised representation $\boldsymbol{s}$ and the concepts $\boldsymbol{c}$ is necessary to ensure that modifying the side channel does not unintentionally alter concept information in the generated outputs.

### 2.1 STATE OF THE ART

Two different approaches have been proposed for implementing the CB module in CBGMs. Ismail et al. (2024) follow a in-hoc approach. They extended the Concept Embedding Model (CEM) (Espinosa Zarlenga et al., 2022) layer by mapping the pre-bottleneck embedding $\boldsymbol{w}$ into an unsupervised embedding and a set of two embeddings per concept, representing the concept's active and inactive states. These embeddings are then combined according to the likelihood of each concept

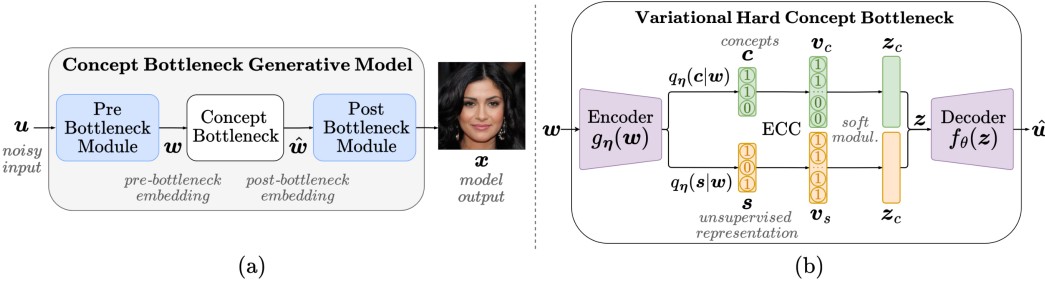

Figure 1: **Block diagram** of (a) the general architecture of CBGMs, and (b) the VHCB layer. Note that the Error-Correcting Code (ECC) in the VHCB layer is a deterministic transformation that enables effective inference.

$c_i$ being active to produce the post-concept embedding $\hat{w}$. The CEM layer and the base generative model are trained jointly.

In contrast, Kulkarni et al. (2025) proposed the Concept Bottleneck Autoencoder (CB-AE), a CB module that can be introduced into pretrained generative models. The encoder maps $w$ into a latent space that includes both concept predictions, with each binary concept represented by two logits, and an unsupervised embedding. The decoder produces the post-concept embedding $\hat{w}$, which must accurately reconstruct $w$ to preserve the pretrained model's generative performance. Training uses synthetic images generated by the pretrained model, with concept labels $y$ obtained from pretrained or zero-shot classifiers such as Contrastive Language-Image Pre-training (CLIP) (Radford et al., 2021). The CB-AE is optimized using

$$\mathcal{L} = \mathcal{L}_{r_1}(w, \hat{w}) + \mathcal{L}_{r_2}(x, \hat{x}) + \mathcal{L}_c(y, c) + \mathcal{L}_{i_1}(y_{\text{target}}, y_{\text{int}}) + \mathcal{L}_{i_2}(y_{\text{target}}, c_{\text{int}}), \qquad (1)$$

where $\mathcal{L}_{r_1}$ and $\mathcal{L}_{r_2}$ are Mean Squared Error (MSE) reconstruction losses on the embeddings $(w, \hat{w})$ and images $(x, \hat{x})$, respectively. $\mathcal{L}_c$ is a cross-entropy loss aligning predicted concepts $c$ with labels $y$. Intervention losses $\mathcal{L}_{i_1}$ and $\mathcal{L}_{i_2}$ enforce that concept interventions produce the desired changes: $\mathcal{L}_{i_1}$ compares target concepts $y_{\text{target}}$ with those in the intervened image $y_{\text{int}}$, while $\mathcal{L}_{i_2}$ aligns the concepts $c_{\text{int}}$ predicted from the intervened embedding $\hat{w}_{\text{int}}$ with $y_{\text{target}}$. Kulkarni et al. (2025) also propose a lightweight version of the CB-AE, referred to as Concept Controller (CC). In this model, only the encoder is trained using a reconstruction loss, and interventions are performed via an optimization-based procedure that applies small perturbations to $w$ to induce a target concept. However, there is no guarantee that these interventions actually utilize the inferred latent concepts, since concept inference and intervention are independent processes. For this reason, we do not consider the CC in our analysis.

**Limitations.** Both methods rely on soft concept representations, which are susceptible to concept leakage, where concept probabilities unintentionally encode task-related information (Havasi et al., 2022; Margeloiu et al., 2021). This reduces control over generation, one of the primary objectives of CBGMs, as models may exploit information shortcuts rather than effectively capture concept information, causing concept interventions to fail to produce the intended effect in the final output. In the case of the CB-AE, this issue is partially mitigated through training-time interventions (see (1)), but at the expense of increased training complexity. Moreover, since both methods are deterministic, they cannot model a generative process over the concept space. As a result, steerable generation is restricted to modifying concepts on existing inputs, while direct sampling from the concept bottleneck or from concept distributions is not supported.

## 3    VARIATIONAL HARD CONCEPT BOTTLENECK LAYER

To address the limitations of existing CBGMs, we propose the Variational Hard Concept Bottleneck (VHCB) layer. The VHCB builds on a binary VAE, in which the binary latent variables that govern the generative process are directly mapped to hard concepts, mitigating concept leakage (Margeloiu et al., 2021; Havasi et al., 2022; Lockhart et al., 2022; Vandenhirtz et al., 2024; Sun et al., 2024). As a result, training can rely on a straightforward reconstruction-based objective without requiring additional intervention terms. Its probabilistic formulation further enables direct generation from

specified concept configurations, while still supporting concept inference and controlled manipulation of generated images through concept interventions. Moreover, the binary latent space yields compact and interpretable concept representations that are easy to manipulate.

We build the VHCB by extending the Coded Discrete Variational Autoencoder (Coded DVAE) from Martínez-García et al. (2025), a state-of-the-art (SOTA) binary VAE, to incorporate two binary latent representations: a vector of concepts $c$, whose components are aligned with predefined hard concepts; and an unsupervised embedding $s$, which acts as a side channel to address concept incompleteness (Ismail et al., 2024; Kulkarni et al., 2025). For each observation $x$, concept supervision is introduced during training using conditional concept distributions $p(y|x)$ as informative priors for $c$. These conditionals $p(y|x)$ are obtained from either a supervised or a zero-shot pretrained classifier, as described in Section 2. We consider the Coded DVAE as base model as it offers (i) a binary latent space with a factorized prior, suitable for modeling binary concepts, (ii) probabilistic estimates of binary latent variables that can be directly mapped to hard concepts, and (iii) effective inference mechanisms that allow for robust modeling of those concepts. Remarkably, despite the discrete latent space, training remains stable through a reparameterization trick based on overlapping smoothing transformations.

### 3.1 MODEL DESCRIPTION

Consider a dataset $\mathcal{D} = (w_i, x_i, y_i)_{i=1}^N$, where $w_i$ is the inner representation of a generative model producing an image $x_i$, and $y_i$ are the associated $K$ binary concept labels. The VHCB maps $w$ to two binary latent representations: a concept vector $c \in \{0,1\}^K$, whose components are aligned with $K$ predefined hard concepts, and an unsupervised representation $s \in \{0,1\}^L$, which serves as a *side channel* to capture generative factors not represented by $c$, accounting for concept incompleteness (Ismail et al., 2024; Kulkarni et al., 2025). We model $c$ and $s$ as binary latent variables with independent Bernoulli components. By defining the side channel as a binary factorized latent space, the model can learn potentially interpretable unsupervised representations, enabling the potential discovery of new concepts.

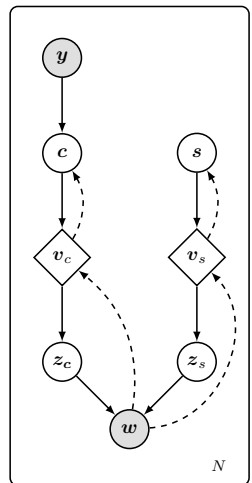

Figure 2: Graphical model of the VHCB.

Prior work showed that *protecting* binary latent variables during generation by introducing structured redundancy with Error-Correcting Codes (ECCs) improves inference in binary VAEs (Martínez-García et al., 2025). An ECC defines a deterministic transformation that increases the dimensionality of the original vector by adding redundancy. In our setting, we protect $c$ and $s$ using two separate ECCs, defining the transformations $c \rightarrow v_c$, with $v_c \in \{0,1\}^{K'}$ and $K' > K$; and $s \rightarrow v_s$, with $v_s \in \{0,1\}^{L'}$ and $L' > L$. These transformations increase the Hamming distance between the binary vectors allowing error correction during inference by selecting the nearest valid codeword in the $K'$-bit space for $c$ and the $L'$-bit space for $s$. Following Martínez-García et al. (2025), we employ uniform repetition codes, where each bit is repeated multiple times to construct the codewords.

Following the Coded DVAE framework, after encoding, a smoothing transformation $p(z|v) = \prod_j p(z_j|v_j)$ is applied to both $v_s$ and $v_c$ to enable differentiable sampling. Specifically, this transformation uses overlapping exponential distributions:

$$p(z|v=1) = \frac{e^{\beta(z-1)}}{Z_\beta}, \quad p(z|v=0) = \frac{e^{-\beta z}}{Z_\beta}, \tag{2}$$

where $v \in \{0,1\}$, $z \in [0,1]$, $Z_\beta = (1 - e^{-\beta})/\beta$, and $\beta$ acts as an inverse temperature parameter. The transformation is applied to both $v_s$ and $v_c$, yielding $z_s$ and $z_c$, which are then concatenated to form the final embedding $z$ that is fed into the decoder Neural Network (NN). The likelihood of the data is conditioned on this smooth auxiliary variable $z$ as $p_\theta(w|z) = p(f_\theta(z))$, where $f_\theta(\cdot)$ denotes the decoder NN.

**Variational Family.** In this setting, the posterior over the hard concepts factorizes as $q_\eta(c, z_c|w) = q_\eta(c|w)p(z_c|v_c)$, while the posterior over the unsupervised representation factorizes analogously

as $q_{\boldsymbol{\eta}}(\boldsymbol{s}, \boldsymbol{z}_s|\boldsymbol{s}) = q_{\boldsymbol{\eta}}(\boldsymbol{s}|\boldsymbol{w})p(\boldsymbol{z}_s|\boldsymbol{v}_s)$. In both cases, we assume independent Bernoulli posteriors, consistent with the Coded DVAE formulation. The marginal posteriors $q_j$, $j = 1, \ldots, K$ of the bits in $\boldsymbol{c}$ and $\boldsymbol{s}$ are obtained in two stages: first, the encoder NN produces the marginals of the coded bits, $q_{\boldsymbol{\eta}}^{\boldsymbol{u}}(\boldsymbol{v}|\boldsymbol{w}) = \prod_{j=1}^{L} \text{Ber}(v_j; g_{j,\boldsymbol{\eta}}(\boldsymbol{w}))$ using the encoder NN $g_{\boldsymbol{\eta}}(\cdot)$. Next, these marginals are refined by leveraging the structure of the repetition code through a *soft majority voting* procedure, yielding improved marginals for the bits in $\boldsymbol{c}$ after correcting errors made by the encoder NN. More details on this process can be found in Martínez-García et al. (2025).

**Training.** The VHCB can be trained either in-hoc, jointly with the base generative model, or post-hoc, using a pretrained model. We focus on the post-hoc setting, as it is more practical: it is more computationally efficient, reduces data requirements, and enables steerable generation with models that already produce high-quality outputs. Following Kulkarni et al. (2025), we generate samples $\boldsymbol{x}_i$ from the pretrained model during training and store their intermediate representations $\boldsymbol{w}_i$. Each generated image $\boldsymbol{x}_i$ is then passed through concept classifiers to obtain $p(\boldsymbol{y}_i|\boldsymbol{x}_i)$, pairing every $\boldsymbol{w}_i$ with the concepts of its generated image. The model is trained by optimizing the following loss

$$\mathcal{L} = \underbrace{\mathbb{E}_{q_{\boldsymbol{\eta}}(\boldsymbol{c},\boldsymbol{s},\boldsymbol{z}|\boldsymbol{w})} \log p_{\boldsymbol{\theta}}(\boldsymbol{w}|\boldsymbol{z})}_{\text{embedding reconstruction}} - \underbrace{\mathcal{D}_{\text{SKL}}\left(q_{\boldsymbol{\eta}}(\boldsymbol{c}|\boldsymbol{w}), p(\boldsymbol{y}|\boldsymbol{x})\right)}_{\text{concept loss}} - \underbrace{\mathcal{D}_{\text{KL}}\left(q_{\boldsymbol{\eta}}(\boldsymbol{s}|\boldsymbol{w})\|p(\boldsymbol{s})\right)}_{\text{side channel regularization}} + \underbrace{\text{MSE}(\boldsymbol{x}, \hat{\boldsymbol{x}})}_{\text{image rec.}},$$

(3)

where $\boldsymbol{z} = [\boldsymbol{z}_c; \boldsymbol{z}_s]$ is the concatenation of the soft latent variables passed to the decoder NN. The first term enforces an accurate reconstruction of the pre-bottleneck embedding $\boldsymbol{w}$. The concept loss aligns the concept posterior $q_{\boldsymbol{\eta}}(\boldsymbol{c}|\boldsymbol{w})$ with $p(\boldsymbol{y}|\boldsymbol{x})$ using the symmetric Kullback-Leibler divergence (*Jeffreys divergence*), defined as $\mathcal{D}_{\text{SKL}}(p, q) \triangleq \mathcal{D}_{\text{KL}}(p\|q) + \mathcal{D}_{\text{KL}}(q\|p)$ (Polyanskiy & Wu, 2025). It preserves distributional modes and prevents the posterior from collapsing to the mean, thereby improving control in concept interventions. We empirically validate this choice through an ablation study available in Appendix D. The side channel regularization term constrains the unsupervised latent $\boldsymbol{s}$ to remain close to its prior. Finally, following Kulkarni et al. (2025), an MSE loss between the original images $\boldsymbol{x}$ and their reconstructions $\hat{\boldsymbol{x}}$ is included. This term is added in the post-hoc setting to preserve the generative quality of the pretrained model, while in in-hoc training it would be replaced by the loss of the underlying generative model (Ismail et al., 2024).

**Steerable generation.** With the VHCB, we can directly sample $\boldsymbol{c}$ and $\boldsymbol{s}$ from its latent space to generate an embedding $\hat{\boldsymbol{w}}$ conditioned on a specified concept distribution. This differs from prior SOTA methods (Ismail et al., 2024; Kulkarni et al., 2025), which only enable steerability through interventions on existing model outputs $\boldsymbol{x}$. Operating directly on hard concepts mitigates leakage and makes interventions transparent and intuitive: controlling a concept reduces to toggling its corresponding bit on (1) or off (0), allowing straightforward manipulation of the generative process. Note that at test time the binary latents $\boldsymbol{c}$ and $\boldsymbol{s}$ can be sampled from a uniform prior or fixed to a given choice, and the smoothing transformation in (2) is applied to obtain $\boldsymbol{z}$. Interventions on generated outputs $\boldsymbol{x}$ are carried out by setting the bit in $\boldsymbol{c}$ corresponding to the target concept to the desired value.

## 4 SYSTEMATIC EVALUATION FOR CBGMS

We aim to design CBGMs that are interpretable, steerable, and capable of producing high-quality outputs. Building on the desiderata outlined in Section 2, we introduce a systematic evaluation framework for CBGMs. We define a set of tasks and metrics that allow to assess performance and analyze important aspects such as concept entanglement, cascading effects of interventions, and the impact of correlations or biases present in the training data or inherited from the pretrained generator.

### 4.1 METRICS

We evaluate model performance using a set of similarity and divergence metrics, selected based on whether concepts are represented as hard variables $\boldsymbol{c} \in \{0, 1\}^K$ or soft class probabilities $p(\boldsymbol{c}|\boldsymbol{w})$. Note that a concept model can be based on hard concepts (those used to generate the datum $\boldsymbol{x}$) and still provide soft class probabilities $q(\boldsymbol{c}|\boldsymbol{w})$. This is the case of the VHCB.

**Metrics for hard concepts.** For hard binary predictions, we evaluate performance using accuracy, defined as $\text{acc}(\boldsymbol{y}, \boldsymbol{c}) = \frac{1}{K} \sum_{j=1}^{K} \mathbf{1}[y_j = c_j]$, where $K$ is the vector dimensionality, $y_j$ is the ground-

truth label, $c_j$ is the predicted concept value, and $\mathbf{1}[\cdot]$ is the indicator function. This metric quantifies the fraction of entries where the predicted vector matches the ground-truth vector.

**Metrics for soft concepts.** Let $\boldsymbol{p}$ be a soft concept vector, where each entry $p_j$ represents the ground-truth class probability $p(y_j = 1|\boldsymbol{x})$, and let $\boldsymbol{q}$ be the corresponding vector of predicted concept probabilities, $q(c_j = 1|\boldsymbol{w})$. In this case, we consider the following metrics:

- *Cosine Similarity.* Quantifies the similarity of two vectors by calculating the cosine of the angle between them, which is given by $\text{sim}(\boldsymbol{p}, \boldsymbol{q}) = \frac{\boldsymbol{p} \cdot \boldsymbol{q}}{\|\boldsymbol{p}\|_2 \|\boldsymbol{q}\|_2}$.

- *Total Variation Distance (TV).* Treating soft binary predictions as Bernoulli distributions allows us to compute statistical distances such as the TV, which quantifies the total change in probability mass between two distributions (Polyanskiy & Wu, 2025). In this case, it is computed per concept and averaged over all $K$ concepts as $\text{TV} = \frac{1}{K} \sum_{j=1}^{K} |p_j - q_j|$.

**Metrics for image generation.** To assess the quality of generated images, we use the standard Fréchet Inception Distance (FID), which quantifies the similarity between the distributions of generated and real images in a learned feature space (Heusel et al., 2017). The FID captures both fidelity and diversity of the generated images.

## 4.2 TASKS

As discussed in Section 2, our goal is to design CBGMs that provide accurate concept inference for interpretability, enable steerable generation through concepts, and preserve high generative quality. In this section, we present a set of tasks to systematically evaluate these properties.

**Robust concept inference.** A primary aspect to evaluate is the model's ability to accurately map pre-bottleneck embeddings $\boldsymbol{w}$ to concepts $\boldsymbol{c}$. Additionally, we need to assess the disentanglement between $\boldsymbol{c}$ and the unsupervised representation $\boldsymbol{s}$, which acts a side channel to address concept incompleteness, ensuring that concept information is captured solely by $\boldsymbol{c}$.

- *Concept prediction.* We assess the model's ability to accurately predict concepts by comparing the concept labels $\boldsymbol{y} \sim p(\boldsymbol{y}|\boldsymbol{x})$ of generated images $\boldsymbol{x}$ with the latent concepts $\boldsymbol{c}$ predicted by the CB layer from their corresponding pre-bottleneck embeddings $\boldsymbol{w}$. In the post-hoc setting, ground-truth labels are obtained from images generated by the base generative model (without the CB layer), and the associated pre-bottleneck embeddings $\boldsymbol{w}$ are mapped to concepts $\boldsymbol{c}$. This ensures that the evaluation reflects how well the model predicts the concepts that are inherently represented in $\boldsymbol{w}$ by the base generator.

- *Disentanglement.* To assess whether the unsupervised representation $\boldsymbol{s}$ encodes concept information, we generate two independent samples from the CBGM, $\boldsymbol{x}_1$ and $\boldsymbol{x}_2$, with corresponding concept labels $\boldsymbol{y}_1, \boldsymbol{y}_2$, latent concept vectors $\boldsymbol{c}_1, \boldsymbol{c}_2$, and unsupervised latents $\boldsymbol{s}_1, \boldsymbol{s}_2$. We then swap the unsupervised latents to form $(\boldsymbol{c}_1, \boldsymbol{s}_2)$ and $(\boldsymbol{c}_2, \boldsymbol{s}_1)$, and produce new outputs $\boldsymbol{x}_1'$ and $\boldsymbol{x}_2'$ with concept labels $\boldsymbol{y}_1'$ and $\boldsymbol{y}_2'$. If $\boldsymbol{s}$ and $\boldsymbol{c}$ are disentangled, concept labels should remain unchanged. We quantify this by computing accuracy between $\boldsymbol{y}_i$ and $\boldsymbol{y}_i'$ and, when soft concept labels are available, by evaluating soft similarity metrics between $p(\boldsymbol{y}|\boldsymbol{x}_i)$ and $p(\boldsymbol{y}'|\boldsymbol{x}_i')$.

**Steerable generation.** A central feature of CBGMs is the ability to steer generation through concepts. To evaluate this, we define tasks that include generation from specified concept configurations and targeted concept interventions in existing generated data. Although CBGMs assume concepts to be independent, they often exhibit statistical relationships (spurious or not) that generative models may capture or even amplify. As a result, random sampling or arbitrary interventions can yield out-of-distribution concept configurations that the model cannot process correctly, limiting steerability. For example, an intervention that activates two mutually exclusive concepts simultaneously, such as 'blond hair' and 'black hair', will fail to generate both. Our evaluation framework accounts for this, allowing us to distinguish errors caused by limitations of the CB layer from those arising from out-of-distribution concept configurations.

- *Direct concept-based generation.* In probabilistic CBGMs, it is possible to sample $(\boldsymbol{c}, \boldsymbol{s})$ from the CB's latent space and generate images conditioned on a specified concept configuration $\boldsymbol{c}$, which can also be out-of-distribution. The sampled pair is mapped to a post-concept embedding

$\hat{w}$, which is then decoded to produce an image $x \sim p(x|\hat{w})p(\hat{w}|s, c)$. Associated concept labels are then estimated with a classifier as $y \sim p(y|x)$. Finally, we measure the accuracy between $y$ and $c$ to evaluate concept alignment.

- *Single concept intervention.* In this case, we evaluate the ability to manipulate individual concepts in existing data. First, a datum $x_0$ is generated with the CBGM, with associated concept vector $c_0$, and concept labels $y_0 \sim p(y_0|x_0)$. The target concept in $c_0$ is then intervened (activated or deactivated) to produce $c_{\text{target}}$. A new output $x_{\text{int}}$ is generated as $x_{\text{int}} \sim p(x_{\text{int}}|\hat{w}_{\text{int}})p(\hat{w}_{\text{int}}|s_0, c_0 \rightarrow c_{\text{target}})$, with corresponding labels $y_{\text{int}} \sim p(y|x_{\text{int}})$. Comparing $y_{\text{int}}$ with $c_{\text{target}}$ and $p(y_0|x_0)$ with $p(y_{\text{int}}|x_{\text{int}})$ allows us to assess (i) whether the intervention is accurately reflected in $x_{\text{int}}$ (Target accuracy) and (ii) whether non-target concepts remain unchanged (Non Target accuracy). To ensure a comprehensive evaluation, interventions are tested in both directions.

- *Minimum Hamming distance intervention.* The evaluation is the same as in the single concept setting, but in this case we target concept patterns that are frequent in the dataset to avoid out-of-distribution configurations. For each sample $x_0$ with latent concept vector $c_0$, we select $c_{\text{target}}$ as the closest training concept configuration in Hamming distance, defined as the number of positions at which two vectors of the same length differ. This approach modifies the fewest possible concepts while staying in-distribution, helping to distinguish errors due to CB layer limitations from those caused by out-of-distribution configurations.

## 5 EXPERIMENTS

In this section, we present experimental results for the proposed VHCB layer [1] and the baseline CB-AE, evaluated on the tasks defined above. As indicated in Section 3, we focus on the post-hoc setting. Experiments are conducted with StyleGAN2 (Karras et al., 2020), pretrained on CelebA-HQ (256×256) (Lee et al., 2020) and CUB-200-2011 (256×256) (Wah et al., 2011), and Denoising Diffusion Probabilistic Model (DDPM) (Ho et al., 2020) pretrained on CelebA-HQ (256×256). Following Kulkarni et al. (2025), for CelebA-HQ we evaluate the model in (i) a large imbalanced regime with all 40 concepts (*all* set), and (ii) a small balanced regime with the 8 most balanced concepts (*balanced* set). In addition, we introduce (iii) a low-correlation subset of 8 balanced concepts that the generator can reliably activate (*low correlation* set), allowing us to isolate the effect of concept correlations. For CUB-200-2011, we adopt the same setup as Kulkarni et al. (2025), using 10 balanced concepts. Results are reported using two pseudo-label sources: supervised ResNet18 classifiers and CLIP zero-shot classifiers. Main results are presented for StyleGAN2 trained on CelebA-HQ and CUB-200-2011, with additional experiments and extended DDPM results reported in the Appendix.

**Model configuration.** For both the CB-AE and VHCB, we use 4-layer Multilayer Perceptrons (MLPs) for the encoder and decoder. In the CB-AE, following Kulkarni et al. (2025), the unsupervised latent is a continuous vector $s \in \mathbb{R}^{40}$. In contrast, the VHCB employs a low-dimensional binary latent $s \in \{0,1\}^5$. Additional implementation details are provided in the Appendix.

**Automated evaluation.** We use independent ResNet50 concept classifiers, not involved in training, to obtain concept labels $y \sim p(y|x)$ for the generated images. These concept labels act as ground truth for evaluating model performance.

**Evaluating single concept interventions.** For each target concept, we identify 1k pre-bottleneck latents $w$ that generate images $x$ with $p(y_{\text{target}} = 1|x) < 0.5$ (concept inactive) and 1k latents with $p(y_{\text{target}} = 1|x) > 0.5$ (concept active). These are then used to do single-concept interventions in both directions, with the same latent set applied across all models to ensure fair comparison.

### 5.1 CONCEPT INFERENCE

As shown in Table 1, the proposed VHCB consistently outperforms the deterministic CB-AE across all concept inference metrics, under both ResNet18 and CLIP supervision. Although both models show reduced performance with CLIP due to the noisier zero-shot labels, VHCB remains superior, demonstrating a more robust mapping from $w$ to $c$. The gains are even larger for disentanglement: VHCB achieves higher accuracy and cosine similarity while maintaining lower TV, indicating that

---

[1]https://github.com/mariamartinezgarcia/vhcb

Table 1: **Concept inference and disentanglement between $c$ and $s$.**

Evaluation on 1k random samples generated by a StyleGAN2 pretrained on CelebA-HQ.

| Pseudo Label Clf. | Model | Concept Set | Concept Inference | | | Disentanglement | | |
|---|---|---|---|---|---|---|---|---|
| | | | Acc. (↑) | Cosine Sim. (↑) | TV(↓) | Acc.(↑) | Cosine Sim. (↑) | TV(↓) |
| ResNet18 | VHCB | all | 0.855 | **0.804** | **0.148** | **0.927** | **0.917** | **0.076** |
| | CB-AE | all | **0.857** | 0.763 | 0.161 | 0.901 | 0.853 | 0.101 |
| | VHCB | low corr. | **0.791** | **0.831** | **0.192** | **0.874** | **0.947** | **0.110** |
| | CB-AE | low corr. | 0.779 | 0.739 | 0.238 | 0.701 | 0.803 | 0.229 |
| CLIP | VHCB | all | **0.623** | **0.613** | **0.337** | **0.921** | **0.901** | **0.083** |
| | CB-AE | all | 0.565 | 0.469 | 0.403 | 0.875 | 0.775 | 0.101 |
| | VHCB | low corr. | 0.599 | **0.730** | **0.299** | **0.854** | **0.943** | **0.125** |
| | CB-AE | low corr. | **0.607** | 0.639 | 0.346 | 0.667 | 0.782 | 0.248 |

|  | $s_1$ | $s_2$ | $s_1$ | $s_2$ | $s_1$ | $s_2$ | $s_1$ | $s_2$ |
|---|---|---|---|---|---|---|---|---|
| $c_1$ | | | | | | | | |
| $c_2$ | | | | | | | | |

Figure 3: **Qualitative evaluation of the disentanglement between $s$ and $c$** with the VHCB layer and StyleGAN2 models pretrained on CelebA-HQ.

changes in $s$ have little effect on generated concepts. The qualitative results shown in Figure 3 further indicate that modifying $s$ does not induce semantic changes in the generated images. This stronger separation results from the compact side channel (5 bits compared to 40 dimensions in CB-AE) of and robust inference mechanisms introduced in VHCB, enforcing a cleaner division between supervised and unsupervised factors.

## 5.2 STEERABLE GENERATION

**Direct generation.** The probabilistic formulation of the VHCB enables direct generation from specified concept configurations. Table 2 reports metrics showing that generated images actually reflect concepts in $c$. Since random sampling of concept configurations can produce out-of-distribution concept configurations, we also sample concept patterns according to their empirical frequency in the training data. This further improves accuracy and highlights how biases in the base generative model can limit the steering capacity of the CBGM. Figure 4 shows examples of generated images with specific concept configurations for both CelebA-HQ and CUB-200-2011.

Table 2: **Steerable Generation.** Evaluation of generation from specific concept configurations using a StyleGAN2 pretrained on CelebA-HQ. Random samples concept sets uniformly at random, while Patterns samples them according to their empirical frequency in the training data.

| Model | Pseudo Label Clf. | Concept Set | Random Target Acc. (↑) | Patterns Target Acc. (↑) |
|---|---|---|---|---|
| VHCB | ResNet18 | all | 0.551 | 0.873 |
| | | low corr. | 0.715 | 0.814 |
| | CLIP | all | 0.533 | 0.830 |
| | | low corr. | 0.632 | 0.687 |

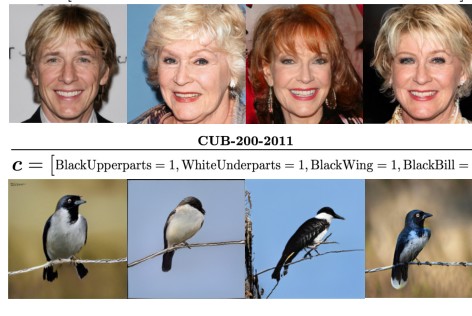

**CelebA-HQ**

$c = \begin{bmatrix} \text{Bangs} = 1, \text{MouthOpen} = 1, \text{Young} = 0, \text{Smiling} = 1, \text{Mustache} = 0 \end{bmatrix}$

**CUB-200-2011**

$c = \begin{bmatrix} \text{BlackUpperparts} = 1, \text{WhiteUnderparts} = 1, \text{BlackWing} = 1, \text{BlackBill} = 1 \end{bmatrix}$

Figure 4: **Steerable generated examples** using the VHCB and a StyleGAN2 pretrained on CelebA-HQ and CUB-200-2011.

Table 3: **Test-time interventions.** Evaluation of single-concept activation $(i \rightarrow a)$, deactivation $(a \rightarrow i)$, and interventions guided by training concept patterns (minimum Hamming distance) using a StyleGAN2 pretrained on CelebA-HQ.

| Pseudo Label Clf. | Model | Concept Set | Single $(i \rightarrow a)$ | | Single $(a \rightarrow i)$ | | Hamming Dist. | |
|---|---|---|---|---|---|---|---|---|
| | | | Target Acc. (↑) | Non Target Acc. (↑) | Target Acc. (↑) | Non Target Acc. (↑) | Target Acc. (↑) | Non Target Acc. (↑) |
| ResNet18 | VHCB | all | **0.170** | 0.903 | **0.550** | 0.902 | **0.660** | 0.938 |
| | CB-AE | all | 0.105 | **0.924** | 0.453 | **0.924** | 0.542 | **0.955** |
| | VHCB | low corr. | **0.769** | 0.762 | **0.765** | 0.781 | **0.634** | 0.845 |
| | CB-AE | low corr. | 0.420 | **0.825** | 0.554 | **0.822** | 0.418 | **0.880** |
| CLIP | VHCB | all | **0.131** | 0.890 | **0.536** | 0.885 | **0.595** | 0.923 |
| | CB-AE | all | 0.085 | **0.922** | 0.311 | **0.922** | 0.466 | **0.949** |
| | VHCB | low corr. | **0.567** | 0.691 | **0.532** | 0.694 | **0.534** | 0.729 |
| | CB-AE | low corr. | 0.317 | **0.829** | 0.404 | **0.834** | 0.496 | **0.869** |

Table 4: **DDPM results**, obtained with a DDPM pretrained on CelebA-HQ.

| Pseudo Label Clf. | Model | Concept Set | Concept Inf. | | Single $(i \rightarrow a)$ | | Single $(a \rightarrow i)$ | |
|---|---|---|---|---|---|---|---|---|
| | | | Acc (↑) | TV (↓) | Target Acc (↑) | Non-Target Acc (↑) | Target Acc (↑) | Non-Target Acc (↑) |
| ResNet18 | VHCB | all | 0.773 | 0.254 | 0.131 | 0.905 | 0.546 | 0.898 |
| | | low corr. | 0.621 | 0.383 | 0.187 | 0.910 | 0.571 | 0.857 |

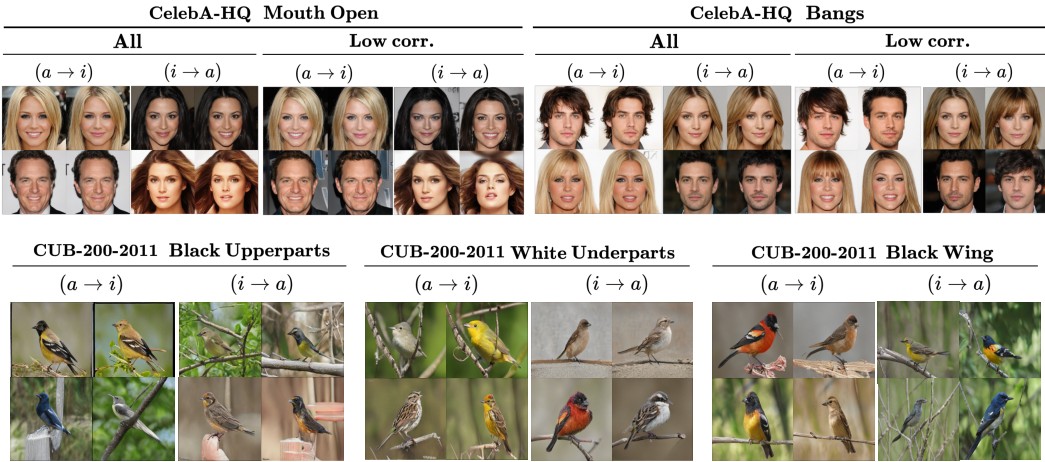

Figure 5: **Single-concept interventions** with the VHCB layer and StyleGAN2 models pretrained on CelebA-HQ and CUB-200-2011.

**Single concept interventions.** We next evaluate the models' ability to manipulate outputs via single-concept interventions, with results reported in Figure 5 and Table 3. The VHCB consistently outperforms the CB-AE in target accuracy, learning a robust mapping from latent concepts $c$ to post-bottleneck embeddings $\hat{w}$ without explicit intervention losses, indicating that hard concept representations enhance steerability. We observe that the CB-AE shows slightly higher non-target accuracy across settings, but this does not indicate more robust interventions. Instead, it reflects a weaker ability to modify the target concept: in many cases, the target is unchanged (lower target accuracy), leaving the output the same and artificially inflating non-target accuracy, which is computed over all 1k interventions. In contrast, the VHCB achieves a better compromise, with large gains in target accuracy (average increase ~46%) and only a minor reduction in non-target accuracy (average drop ~7%). For example, on the low-correlation set with ResNet18, target accuracy rises from

0.42 to 0.77, while non-target accuracy decreases slightly from 0.83 to 0.76. We provide qualitative examples of this behavior in the Appendix, for both VHCB and CB-AE.

We also find that steerability is strongly influenced by the concept distribution in the dataset. In CelebA-HQ, most concepts are underrepresented, making deactivation easier than activation, as reflected by higher success rates for deactivating concepts. The effects of imbalance and model biases are evident when comparing the full set of 40 unbalanced concepts to balanced or low-correlation subsets, where performance metrics consistently improve for both models (results for the balanced set are provided in the Appendix). Thus, the composition of the concept set has a significant impact on CBGM performance. We observe the same effect in the DDPM results in Table 4. Additional results illustrating the effects of concept incompleteness and biases are provided in the Appendix.

**Hamming distance interventions.** Arbitrary single-concept interventions can produce out-of-distribution concept configurations that the model cannot generate correctly, resulting in failed interventions. To mitigate this, we restrict interventions to target one of the 100 most probable concept patterns observed in the dataset, selected via minimum Hamming distance. This approach consistently improves intervention success rates in both models, with the VHCB still yielding superior metrics, as shown in Table 3. These results further support that steerability depends not only on the expressiveness of the CB layer but also on biases inherent in the pretrained generative model.

## 6 CONCLUSION

In this work, we introduced the VHCB layer, the first probabilistic CBGM with hard concepts. The VHCB enables improved steerability by (i) mitigating concept leakage by considering hard concepts, and (ii) enabling direct generation from specified concept configurations thanks to its probabilistic formulation, while still supporting concept inference and steerability through concept interventions. We also proposed a systematic evaluation framework for CBGMs, through which we showed that the VHCB consistently enhances steerability. This framework further revealed the impact of correlations and biases in training data, which are not captured under our assumption of independent concepts. Future work includes modeling concept relationships in the latent space and fine-tuning base models to mitigate spurious biases exposed through steerability analyses.

## 7 ACKNOWLEDGMENTS

We would like to thank Sofia Molotkova for her support in early stages of the project and every member of the Probabilistic Machine Learning group (UdS) for their invaluable feedback, as well as the anonymous reviewers whose feedback helped us improve this manuscript. This work has been funded by the European Union (ERC-2021-STG, SAML, 101040177, PI Valera) and (MSCA-2024-DNS, MLCARE, 101226456, PI Olmos). However, the views and opinions expressed are those of the author(s) only and do not necessarily reflect those of the European Union or the European Research Council Executive Agency. Neither the European Union nor the granting authority can be held responsible.

Alejandro Lancho and Ricardo Vazquez have received funding from the Comunidad de Madrid's 2023 Cesar Nombela program under Grant Agreement No. 2023-T1/COM-29065. Additionally, Alejandro Lancho has received funding from the Ministerio de Ciencia, Innovación y Universidades, Spain, under Grant Agreement No. PID2023-148856OA-I00 (funded by MICIU/AEI/ 10.13039/501100011033 and by ERDF/UE).

Pablo M. Olmos has also been supported by the Comunidad de Madrid IND2024/TIC-34728, IDEA-CM project (TEC-2024/COM-89), the ELLIS Unit Madrid (European Laboratory for Learning and Intelligent Systems), the 2024 Leonardo Grant for Scientific Research and Cultural Creation from the BBVA Foundation, and from the Ministerio de Ciencia, Innovación y Universidades, Spain, under Grant Agreements No. PID2024- 157856NB-I00 CARTESIAN,PID2021-123182OB-I00 EPi-CENTER (funded by MICIU/AEI/ 10.13039/501100011033 and by ERDF/UE).

## 8 REPRODUCIBILITY STATEMENT

We have made a strong effort to ensure the reproducibility of our results by providing detailed descriptions of the implementation, training procedures, and evaluation framework. In particular, we explain how the method integrates with different generative architectures, including StyleGAN2 and DDPM, specify the pretrained models used in our experiments, detail the training procedures for each architecture, and describe the concept sets employed for the various model configurations (with a brief overview in the main text and a more extensive discussion in the Appendix). Additionally, we provide a comprehensive description of the evaluation framework, including implementation details and dataset-specific considerations.

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

## A    EXTENDED BACKGROUND

**Interpretability in Generative Models.** A central research direction focuses on learning disentangled representations, where independent factors of variation are captured along latent dimensions to enable controlled generation (Bengio et al., 2013; Chen et al., 2016; Higgins et al., 2017; Paige et al., 2017; Jeon et al., 2021; Yang et al., 2023). Although these factors can sometimes be interpretable, disentanglement does not guarantee alignment with human-interpretable concepts (Locatello et al., 2019). Also post-hoc approaches have been proposed, such as Adel et al. (2018), where an existing latent space is transformed using invertible transformations to encode interpretable concepts. However, interventions in this setting are not direct as more than one latent dimension can map to the same concept information. Another method aims to identify interpretable control directions in Generative Adversarial Networks (GANs) by applying Principal Component Analysis (PCA) to internal representations of the model (Härkönen et al., 2020). A different line of work, mainly applied in Natural Language Processing (NLP), trains classifiers to predict concepts from model representations (Alain & Bengio, 2017; Belinkov, 2022). While this reveals that internal states contain concept information, it does not imply that the model relies on such concepts for its predictions.

**Concept Bottleneck Models (CBMs).** A CBM (Koh et al., 2020) is an inherently interpretable model consisting of two modules: a concept predictor, which maps inputs to human-understandable concepts, and a label predictor, which maps these concepts to task outputs. These modules can be trained independently, sequentially, or jointly, and concepts may be represented as discrete labels, class probabilities, or embeddings (Koh et al., 2020; Havasi et al., 2022; Espinosa Zarlenga et al., 2022). CBMs offer two key advantages: (i) predictions are explicitly grounded in interpretable high-level concepts, and (ii) users can intervene on these concepts to alter the final output. However, they face two main challenges: concept incompleteness and concept leakage. Incompleteness arises when the predefined concept set fails to capture all task-relevant information, which can be mitigated through a *side channel* or by learning representations that encode additional information not explained by the concepts (Havasi et al., 2022; Espinosa Zarlenga et al., 2022; Sawada & Nakamura, 2022; Yuksekgonul et al., 2023). Leakage occurs when soft concept probabilities unintentionally encode task labels, undermining interpretability and trustworthiness (Margeloiu et al., 2021; Havasi et al., 2022). This can be alleviated by using binary concept representations or enforcing uncorrelated concept representations (Chen et al., 2020; Margeloiu et al., 2021; Havasi et al., 2022; Espinosa Zarlenga et al., 2022; Marconato et al., 2022; Sheth & Ebrahimi Kahou, 2023; Vandenhirtz et al., 2024). In our work, we adopt hard binary concepts leveraging the Coded DVAE, which prevent leakage while providing a low-dimensional and easily interpretable latent representation.

### A.1    RELATED WORK: INTERPRETABILITY IN GENERATIVE MODELS THROUGH CBMS

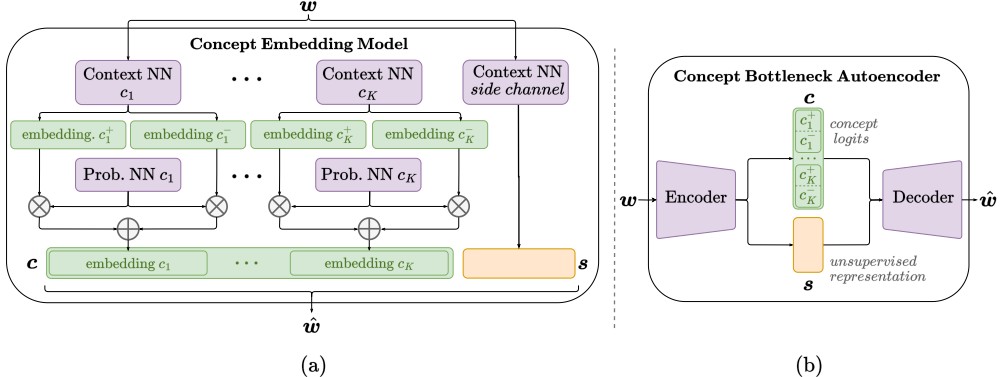

Figure 6: Diagrams of SOTA CBGM architectures: (a) the Concept Embedding Model (CEM) (Ismail et al., 2024) and the (b) Concept Bottleneck Autoencoder (CB-AE) (Kulkarni et al., 2025).

**Concept Embedding Model (CEM).** Ismail et al. (2024) were the first to introduce CBMs into generative models, aiming to make them both interpretable and steerable through a model-agnostic concept bottleneck layer. Building on the CEM layer (Espinosa Zarlenga et al., 2022), the pre-

bottleneck representation $w$ is mapped into two embeddings per concept, representing its active and inactive states, which are mixed based on the likelihood of the concept $y_i$ being active. To capture task-relevant information not covered by predefined concepts, the authors introduce an unknown concept network that learns an additional unsupervised embedding. In this framework, the generative model and bottleneck layer must be trained jointly, making the approach reliant on annotated data and computationally expensive.

**Post-Hoc Concept Bottleneck Autoencoder (CB-AE) and CC.** To reduce the dependence on labeled data and the computational cost of CBGMs, Kulkarni et al. (2025) proposed the CB-AE, a concept-based autoencoder that can be integrated into pretrained generative models. The encoder maps the pre-bottleneck latent $w$ into a concept space that includes both concept predictions, with each binary concept represented by two logits, and an unsupervised embedding that captures information not covered by the concepts. The decoder reconstructs the latent $\hat{w}$, which is passed to the post-bottleneck module to generate the final output $\hat{x}$. In this setup, depicted in Figure 6, only the CB-AE is trained, while the generative model remains fixed. To this end, the CB-AE leverages synthetic images generated by the pretrained model, with pseudo-labels $y$ given by pretrained or zero-shot classifiers such as CLIP (Radford et al., 2021). The model is trained by minimizing the following objective

$$\mathcal{L} = \mathcal{L}_{r_1}(w, \hat{w}) + \mathcal{L}_{r_2}(x, \hat{x}) + \mathcal{L}_c(y, c) + \mathcal{L}_{i_1}(y_{\text{target}}, y_{\text{int}}) + \mathcal{L}_{i_2}(y_{\text{target}}, c_{\text{int}}), \qquad (4)$$

where $\mathcal{L}_{r_1}$ and $\mathcal{L}_{r_2}$ are MSE reconstruction losses between generative latents $(w, \hat{w})$ and images $(x, \hat{x})$. $\mathcal{L}_c$ is a cross-entropy loss aligning predicted concepts $c$ with pseudo-labels $y$. The intervention losses $\mathcal{L}_{i_1}$ and $\mathcal{L}_{i_2}$ are cross-entropy losses that enforce that interventions in the concept space yield the desired changes in both the generative latent $\hat{w}_{\text{int}}$ and the generated image $\hat{x}_{\text{int}}$: $\mathcal{L}_{i_1}$ compares the target concepts $y_{\text{target}}$ with the pseudo-labels of the intervened image $y_{\text{int}}$, while $\mathcal{L}_{i_2}$ aligns the predictions $c_{\text{int}}$, obtained by mapping $\hat{w}_{\text{int}}$ to the concept space, with $y_{\text{target}}$. In this case, interventions are performed by swapping the concept logits.

Alternatively, the authors propose optimization-based interventions where small perturbations are applied to the latent $w$ to induce a target concept, an approach referred to as CC. In this setting, only the encoder is trained with reconstruction losses, but there is no guarantee that the model relies on latent concept information to carry out the interventions.

## B  STYLEGAN2

The main experimental results of this work were obtained by integrating our proposed VHCB layer into pretrained StyleGAN2 models (Karras et al., 2020). This family of generative models is particularly well suited to the CBGM framework, as image generation is controlled by a single latent representation that we map to concepts and that the generator then transforms into an output. In StyleGAN2, we place the VHCB layer between the Mapping Network and the Synthesis Network, as illustrated in Figure 7, following the approach of Ismail et al. (2024) and Kulkarni et al. (2025). These prior works showed that this the optimal location in this setting, since it enables high-quality image generation while allowing the CB layer to capture semantic rather than style information.

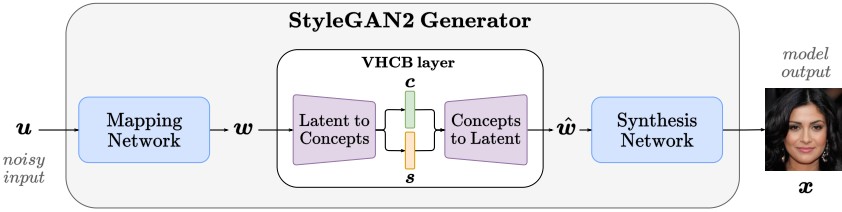

Figure 7: Diagram showing the integration of our VHCB layer into the StyleGAN2 architecture.

**Datasets and pretrained models.** We use two publicly available pretrained StyleGAN2 models: one trained on CelebA-HQ at $256 \times 256$ resolution (Lee et al., 2020), and one trained on CUB 200-2011 at $256 \times 256$ resolution (Wah et al., 2011)[2]. Following Kulkarni et al. (2025), for CelebA-HQ

---

[2]https://github.com/NVlabs/stylegan3

we train and evaluate the VHCB layer in (i) a large imbalanced regime with all 40 concepts (*all* set), and (ii) a small balanced regime with the 8 most balanced concepts (*balanced* set). In addition, we introduce (iii) a low-correlation subset of 8 balanced concepts that the generator can reliably activate (*low correlation* set), allowing us to isolate the effect of concept correlations. For CUB, we adopt the same setup as Kulkarni et al. (2025), using 10 balanced concepts. The specific concept sets are as follows:

- **CelebA-HQ**
    - *Complete set (all, 40 concepts):* '5 o Clock Shadow', 'Arched Eyebrows', 'Attractive', 'Bags Under Eyes', 'Bald', 'Bangs', 'Big Lips', 'Big Nose', 'Black Hair', 'Blond Hair', 'Blurry', 'Brown Hair', 'Bushy Eyebrows', 'Chubby', 'Double Chin', 'Eyeglasses', 'Goatee', 'Gray Hair', 'Heavy Makeup', 'High Cheekbones', 'Male', 'Mouth Slightly Open', 'Mustache', 'Narrow Eyes', 'No Beard', 'Oval Face', 'Pale Skin', 'Pointy Nose', 'Receding Hairline', 'Rosy Cheeks', 'Sideburns', 'Smiling', 'Straight Hair', 'Wavy Hair', 'Wearing Earrings', 'Wearing Hat', 'Wearing Lipstick', 'Wearing Necklace', 'Wearing Necktie', and 'Young'.
    - *Balanced set (8 concepts):* 'Attractive', 'Wearing Lipstick', 'Mouth Slightly Open', 'Smiling', 'High Cheekbones', 'Heavy Makeup', 'Male', and 'Arched Eyebrows'.
    - *Low correlation set (8 concepts):* 'Pale Skin', 'Bangs', 'Big Lips', 'Mouth Slightly Open', 'Chubby', 'Young', 'Smiling', and 'Mustache'.

- **CUB-200-2011**
    - *Balanced set (10 concepts):* 'Small size 5, to 9 inches', 'Perching like shape', 'Solid breast pattern', 'Black bill color', 'Bill length shorter than head', 'Black wing color', 'Solid belly pattern', 'All purpose bill shape', 'Black upperparts color', and 'White underparts color'.

**Implementation and training details.** For both the CB-AE and VHCB, we use 4-layer MLPs for the encoder and decoder. In the CB-AE, following Kulkarni et al. (2025), the unsupervised latent is a continuous vector $s \in \mathbb{R}^{40}$. In contrast, the VHCB employs a low-dimensional binary latent $s \in \{0,1\}^5$, protected by uniform repetition with a code rate of $R = 5/50$. Concept variables are encoded with code rates $R \in \{8/240, 10/300, 40/800\}$ depending on the number of concepts. Models are trained for 50 epochs using Adam with a learning rate of $0.0002$, with concept labels provided either by supervised ResNet18 classifiers or CLIP zero-shot classifiers. The VHCB used batch size 32 and 1000 steps per epoch, and weighted the concept loss by $\beta = 20$ in the training objective. For the CB-AE, we halved the batch size to 16 due to memory limits and doubled the steps to match the number of training samples. For CelebA-HQ with the balanced concept set, we used the model weights provided by the authors [3]; all other configurations were trained from scratch. Thresholds for computing the concept losses in the CB-AE were set to $0.9$ for supervised classifiers, $0.6$ for CelebA-HQ with CLIP, and none for CUB-200-2011, following the original code specifications.

## C  DENOISING DIFFUSION PROBABILISTIC MODEL

To assess the generality of the proposed VHCB across generative architectures, we integrate it into a DDPM (Ho et al., 2020). Following the setup of Kulkarni et al. (2025), we use a publicly available pretrained DDPM from Google, trained on CelebA-HQ at $256 \times 256$ resolution, which employs a U-Net as the denoising network.[4] This enables a direct comparison with the baseline method from Kulkarni et al. (2025) under identical conditions.

We have considered two different implementations. In Section C.1, following Kulkarni et al. (2025), we attach the VHCB module to the bottleneck layer of the U-Net architecture. In Section C.2, we attach the output of the VHCB module to all levels of the DDPM's U-Net architecture in the upsampling path.

---

[3]https://github.com/Trustworthy-ML-Lab/posthoc-generative-cbm
[4]https://huggingface.co/google/ddpm-celebahq-256

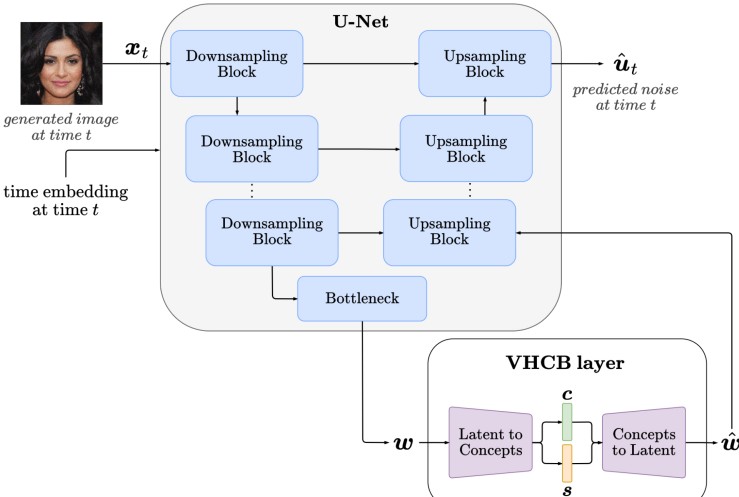

Figure 8: Integration of the VHCB module into the pretrained DDPM architecture. A noisy image $\mathbf{x}_t$ at diffusion timestep $t$ is processed by the U-Net, which predicts the corresponding noise $\hat{\epsilon}_t$. The VHCB module is inserted at the bottleneck of the U-Net and maps intermediate features to a binary concept vector $\mathbf{c}$ via an encoder, which is then used for concept prediction, steerability, and reconstruction via the decoder. The objective is to ensure that semantic information is captured and can be manipulated through $\mathbf{c}$, without disrupting the denoising trajectory.

## C.1 VHCB AT BOTTLENECK

Our aim here is to replicate the original post-hoc CBM configuration and verify whether the trends observed with StyleGAN2 also hold in diffusion models. A schematic of the model with the VHCB module is shown in Figure 8.

**Implementation and training details.** We follow the same dataset and concept setup as in the StyleGAN2 experiments (see Section B), but restrict our analysis to CelebA-HQ. We first generate a dataset of 50,000 images using the pretrained DDPM, storing both the latent noise vectors and the resulting images. Concept pseudo-labels are obtained using a bank of ResNet-18 classifiers (one per concept), and probabilities are thresholded at 0.5 to produce binary targets.

The VHCB is trained by replaying the diffusion process: the stored noise latents are fed into the fixed pretrained DDPM, now augmented with our module. We restrict training to timesteps $t \in [0, 400]$, since later steps add excessive noise and degrade the semantic structure in the bottleneck representation (Kulkarni et al., 2025). At each step, the U-Net predicts the noise at timestep $t$, and the VHCB receives the corresponding bottleneck features.

The loss is the same supervised objective as in equation (3), consisting of (i) a binary cross-entropy loss for concept prediction, (ii) a KL divergence regularizer for the side channel, and (iii) a reconstruction term. In the DDPM setting, the reconstruction is implemented as a consistency constraint: the mean squared error between the predicted noise of the pretrained U-Net and that of the U-Net with the VHCB attached. This ensures that the VHCB learns concept-aligned representations without interfering with the denoising process.

We use the same VHCB architecture described in Section B. However, here we only consider code rates $R \in \{8/240, 40/800\}$ depending on the number of concepts. Models are trained for 50 epochs using Adam with a learning rate of 0.0002, batch size of 32, with concept labels provided either by supervised ResNet18 classifiers or CLIP zero-shot classifier. A weight of $\beta = 20$ was again applied to the concept loss term to increase its influence during training.

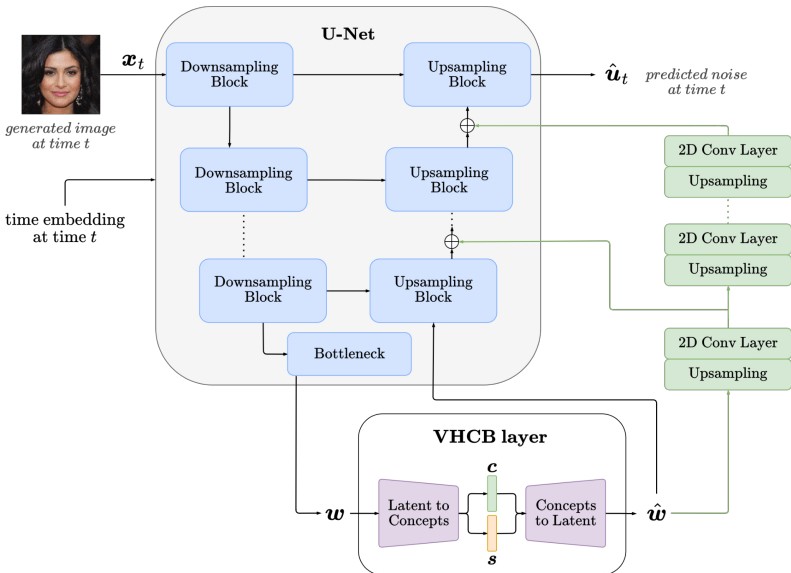

Figure 9: DDPM architecture with VHCB at all levels, featuring a separate decoder-like network that adapts the VHCB bottleneck feature resolution to match each decoder output layer in the U-Net's upsampling path (shown in green at the right side of the figure). The $\oplus$ symbol indicates element-wise addition of the corresponding outputs.

## C.2 DDPM EXTENSION: VHCB AT ALL LEVELS

Following the same procedure as the original DDPM implementation, where the VHCB layer was placed only after the bottleneck, we now attach the output of the VHCB module to all levels of the DDPM's U-Net architecture in the upsampling path.

**Implementation and training details**. This extension introduces a dedicated upsampling path (decoder like architecture) that takes the VHCB's reconstructed bottleneck representation and progressively transforms it through a series of upsampling and 2D convolutional layers. At each stage, the network expands and refines the bottleneck features so they match the required spatial resolution of each decoder output layer in the upsampling path of the U-Net. These adapted bottleneck features are then added element-wise to the outputs of the respective decoder layers. A schematic overview of this extension is provided in Figure 9.

The training procedure for this extension follows the same setup as the original implementation described in Section C.1, with a few additional considerations. To reduce overall training time, the VHCB layer was initialized using the weights from its previously trained model (VHCB-at-bottleneck). An early stopping strategy was also employed, becoming active only after the first 20 epochs and terminates training if no improvement is observed within a patience window of 5 epochs. The maximum number of epochs is set to 50, matching the previous training configuration. The consistency term, previously defined as the mean squared error between the predicted noise of the pretrained U-Net and that of the U-Net with the VHCB attached, is now computed between the predicted noise of the U-Net without the VHCB extension and the predicted noise obtained when the bottleneck-adjusted representation is injected into each decoder output in the upsampling path. The same learning rate and batch size were also used.

## D ABLATION STUDIES

We have conducted some ablation studies to gain intuition on the behavior of the model under different configurations, and empirically validate some design choices.

**Ablation study on the concept loss.** To empirically validate the choice of the symmetric KL divergence as concept loss, we have conducted an ablation study comparing thethe performance on

Table 5: **Ablation on the concept loss.** StyleGAN2 CelebA-HQ.

| Concept Set | Pseudo Label Clf. | Concept Loss | Target Acc. (↑) $(i \to a)$ | Target Acc. (↑) $(a \to i)$ | Target Acc. (↑) *Hamming Distance* | FID (↓) |
|---|---|---|---|---|---|---|
| all | ResNet18 | BCE | 0.165 | 0.555 | 0.676 | 0.543 |
| | ResNet18 | KL | 0.168 | 0.524 | 0.669 | 0.546 |
| | ResNet18 | Symm. KL | 0.170 | 0.550 | 0.660 | 0.551 |
| | CLIP | BCE | 0.127 | 0.564 | 0.621 | 0.527 |
| | CLIP | KL | 0.128 | 0.508 | 0.603 | 0.530 |
| | CLIP | Symm. KL | 0.131 | 0.536 | 0.594 | 0.533 |
| balanced | ResNet18 | BCE | 0.395 | 0.564 | 0.527 | 0.632 |
| | ResNet18 | KL | 0.314 | 0.748 | 0.726 | 0.663 |
| | ResNet18 | Symm. KL | 0.375 | 0.814 | 0.594 | 0.670 |
| | CLIP | BCE | 0.405 | 0.654 | 0.721 | 0.613 |
| | CLIP | KL | 0.306 | 0.745 | 0.767 | 0.602 |
| | CLIP | Symm. KL | 0.346 | 0.682 | 0.423 | 0.607 |

Table 6: **Ablation on the side channel regularization.** Concept inference and steerability results on StyleGAN2 CelebA-HQ, using supervised ResNet18 classifiers for concept supervision during training.

| Concept Set | KL Weight | Size $s$ | Concept Acc. (↑) *(inference)* | Target Acc. (↑) $(i \to a)$ | Target Acc. (↑) $(a \to i)$ | Target Acc. (↑) *(Hamming Dist.)* |
|---|---|---|---|---|---|---|
| all | 0 | 5 | 0.853 | 0.173 | 0.558 | 0.666 |
| | 1 | 5 | 0.855 | 0.170 | 0.550 | 0.660 |
| | 10 | 5 | 0.856 | 0.173 | 0.562 | 0.656 |
| | 20 | 5 | 0.853 | 0.173 | 0.544 | 0.673 |
| | 40 | 5 | 0.850 | 0.174 | 0.550 | 0.659 |
| balanced | 0 | 5 | 0.773 | 0.358 | 0.784 | 0.627 |
| | 1 | 5 | 0.752 | 0.375 | 0.814 | 0.594 |
| | 10 | 5 | 0.754 | 0.362 | 0.822 | 0.609 |
| | 20 | 5 | 0.763 | 0.364 | 0.817 | 0.596 |
| | 40 | 5 | 0.763 | 0.337 | 0.799 | 0.561 |
| balanced | 0 | 10 | 0.766 | 0.363 | 0.785 | 0.606 |
| | 1 | 10 | 0.770 | 0.356 | 0.788 | 0.514 |
| | 10 | 10 | 0.774 | 0.351 | 0.822 | 0.509 |
| | 20 | 10 | 0.774 | 0.350 | 0.818 | 0.499 |
| | 40 | 10 | 0.774 | 0.326 | 0.811 | 0.500 |

steerability and generation of models trained with Binary Cross Entropy (BCE), forward KL, and symmetric KL as concept loss. These results complement our provided intuition in Section 3.1, by illustrating that models trained with BCE tend to perform slightly better when *deactivating* concepts, models trained with forward KL generally achieve slightly better control when *activating* concepts, and models trained with symmetric KL provide the best balance, offering strong interventional performance in both directions.

**Ablation on the side channel regularization.** To determine the impact of the side channel on the model's overall behavior, we conducted an ablation on the side-channel regularization term in the loss function (Eq. (3)) by varying the unsupervised KL weight for various side-channel dimensions and concept sets. The obtained results show that this parameter does not have any significant effect in any of the reported metrics when the concept set is large (i.e., 40 concepts in CelebA-HQ). When considering small concept sets, both steerability and FID slightly deteriorate as side-channel regularization increases. This can be explained by the fact that the proposed CBGM is a generative model optimized to minimize reconstruction loss. When only a small subset of concepts is used and the side channel is over-regularized the expressivity of the model is constrained, which in turn limits both the quality of generated samples (measured by FID) and, consequently, the steerability.

Table 7: **Ablation on the side channel regularization.** Generation results on StyleGAN2 CelebA-HQ, using supervised ResNet18 classifiers for concept supervision during training.

| Concept Set | KL Weight | Size $s$ | Target Acc. ($\uparrow$) (generation) | FID ($\downarrow$) (forward) |
|---|---|---|---|---|
| all | 0 | 5 | 0.548 | 6.855 |
| | 1 | 5 | 0.551 | 7.248 |
| | 10 | 5 | 0.550 | 7.546 |
| | 20 | 5 | 0.549 | 7.590 |
| | 40 | 5 | 0.543 | 7.743 |
| balanced | 0 | 5 | 0.659 | 11.130 |
| | 1 | 5 | 0.670 | 11.019 |
| | 10 | 5 | 0.609 | 11.870 |
| | 20 | 5 | 0.596 | 12.374 |
| | 40 | 5 | 0.561 | 12.687 |
| balanced | 0 | 10 | 0.669 | 10.946 |
| | 1 | 10 | 0.773 | 8.661 |
| | 10 | 10 | 0.663 | 10.309 |
| | 20 | 10 | 0.664 | 10.481 |
| | 40 | 10 | 0.665 | 9.216 |

Table 8: **Ablation on the side channel regularization.** Disentanglement results on StyleGAN2 CelebA-HQ, using supervised ResNet18 classifiers for concept supervision during training.

| Concept Set | KL Weight | Size $s$ | DCI $c$ ($\uparrow$) | DCI $s$ ($\uparrow$) | DCI $[c, s]$ ($\uparrow$) | Acc. $c$ ($\uparrow$) (after swap) | Acc. $s$ ($\uparrow$) (after swap) |
|---|---|---|---|---|---|---|---|
| all | 0 | 5 | 0.163 | 0.095 | 0.161 | 0.984 | 0.838 |
| | 1 | 5 | 0.156 | 0.103 | 0.154 | 0.983 | 0.822 |
| | 10 | 5 | 0.162 | 0.107 | 0.160 | 0.982 | 0.796 |
| | 20 | 5 | 0.149 | 0.098 | 0.148 | 0.980 | 0.750 |
| | 40 | 5 | 0.153 | 0.087 | 0.151 | 0.980 | 0.677 |
| balanced | 0 | 5 | 0.206 | 0.096 | 0.202 | 0.994 | 0.805 |
| | 1 | 5 | 0.207 | 0.043 | 0.201 | 0.999 | 0.798 |
| | 10 | 5 | 0.198 | 0.086 | 0.192 | 0.997 | 0.813 |
| | 20 | 5 | 0.194 | 0.076 | 0.188 | 0.998 | 0.755 |
| | 40 | 5 | 0.180 | 0.073 | 0.176 | 0.994 | 0.712 |
| balanced | 0 | 10 | 0.260 | 0.091 | 0.235 | 0.989 | 0.819 |
| | 1 | 10 | 0.249 | 0.090 | 0.240 | 0.990 | 0.819 |
| | 10 | 10 | 0.230 | 0.071 | 0.213 | 0.983 | 0.742 |
| | 20 | 10 | 0.228 | 0.074 | 0.214 | 0.983 | 0.742 |
| | 40 | 10 | 0.212 | 0.084 | 0.210 | 0.983 | 0.696 |

Table 9: **Effect of removing the unsupervised latent $s$.** Disentanglement results on StyleGAN2 CelebA-HQ, using supervised ResNet18 classifiers for concept supervision during training.

| Concept Set | Size $s$ | Target Acc. ($\uparrow$) ($i \rightarrow a$) | Target Acc. ($\uparrow$) ($a \rightarrow i$) | Target Acc. ($\uparrow$) (Hamming Dist.) | Target Acc. ($\uparrow$) (generation) | FID ($\downarrow$) (forward) |
|---|---|---|---|---|---|---|
| all | 5 | 0.170 | 0.550 | 0.660 | 0.551 | 7.248 |
| | 0 | 0.191 | 0.538 | 0.664 | 0.539 | 7.921 |
| balanced | 5 | 0.375 | 0.810 | 0.594 | 0.670 | 11.016 |
| | 0 | 0.329 | 0.711 | 0.420 | 0.625 | 20.950 |
| low corr. | 5 | 0.769 | 0.765 | 0.634 | 0.715 | 11.589 |
| | 0 | 0.604 | 0.837 | 0.765 | 0.688 | 19.501 |

We also analyzed the disentanglement between $c$ and $s$ for different levels of regularization of the side channel. For this, we use the Disentanglement, Completeness, and Informativenes (DCI) metric

(Carbonneau et al., 2022) and replicate the disentanglement experiment using the autoencoder architecture of the VHCB layer. The procedure follows the same steps as the original disentanglement experiment explained in Section 4.2, with the only difference that, after swapping the unsupervised vectors $s_1$ and $s_2$, we project the reconstructed latent representation back into the VHCB latent space. If $c$ and $s$ are indeed disentangled, both components should be accurately recovered after the swap. In the case of the DCI we observe that: (i) the DCI for $s$ is always significantly lower that the DCI for $c$, indicating that concepts are not directly encoded in the unsupervised latent; (ii) the DCI for $c$ and for $[c, s]$ is comparable, suggesting that $s$ does not contain relevant concept information. Similar to the steerability results, we observe a slight degradation in these metrics, likely caused by the limited expressivity of the model. Nevertheless, the disentanglement experiment (*accuracy after swap*) shows that the concept information can still be reliably recovered even after modifying the side channel, regardless of the degree of regularization applied to it.

**Effect of removing the unsupervised latent $s$.** The unsupervised latent $s$ (side channel) is used to handle concept incompleteness. As discussed in Section 3.1, generation depends on factors that cannot be fully captured by human-interpretable concepts (e.g., texture, lighting, or object position), making the predefined concept set inherently incomplete. Removing the side channel produces the following effects: (i) for large concept sets, we observe that removing the side channel does not have a significant impact in performance, following the results obtained in the ablation of the weight of the side channel regularization; (ii) for reduced concept sets, we observe a significant drop in generation quality, with FID nearly doubling, due to the limited expressiveness of the latent space; (iii) removing the side channel significantly reduces the generative model's expressivity and, consequently, also (slightly) its steerability.

# E    ADDITIONAL EXPERIMENTAL RESULTS

In this section, we extend the experiments from the main paper. Specifically, we report results using pretrained StyleGAN2 models for the three CelebA-HQ concept sets, as well as results for the CUB-200-2011 dataset. We also expand the experimental evaluation by including results obtained with a pretrained DDPM.

**Intervening concepts.** In the VHCB, test-time interventions are performed by setting the latent variable of the target concept to the desired value (0 for inactive, 1 for active). For the CB-AE, interventions are applied by assigning the maximum logit (out of the two representing each concept) to the position corresponding to the target value. Metrics for single-concept interventions are calculated by systematically intervening on every concept in the set, with the aim of changing the target concept while keeping all other concepts unchanged.

**Evaluating single concept interventions.** For each target concept, we identify 1k pre-bottleneck latents $w$ that generate images $x$ with $p(y_{\text{target}} = 1|x) < 0.5$ (target concept inactive) and 1k latents with $p(y_{\text{target}} = 1|x) > 0.5$ (target concept active). These latents are then used to do single-concept interventions in both directions, with the same latent set applied across all models to ensure fair comparison. Because some concepts are underrepresented in the dataset, the pretrained generative model may fail to reliably generate images where those concepts are active. As a result, in the large imbalanced CelebA-HQ regime, 1k active latents could only be obtained for a subset of the 40 concepts. Specifically, active latents were obtained for the following concepts: 'Arched Eyebrows', 'Attractive', 'Bangs', 'Big Lips', 'Big Nose', 'Chubby', 'Goatee', 'Heavy Makeup', 'High Cheeckbones', 'Male', 'Mouth Slightly Open', 'Mustache', 'No Beard', 'Pale Skin', 'Smiling', 'Wearing Lipstick', and 'Young'.

## E.1    STYLEGAN2 ON CELEBA-HQ

**Concept inference.** We observe the results generalize across the 3 considered concept subsets. As shown in Table 10, our proposed VHCB consistently improves over the deterministic CB-AE across concept inference metrics, under both ResNet18 and CLIP pseudo-label supervision. Notably, the VHCB improves the soft metrics, suggesting that its posterior is properly aligned the pretrained classifiers. While performance decreases for both models with CLIP supervision due to the noisier zero-shot labels, VHCB remains clearly superior, indicating a more robust mapping from $w$ to $c$. The gains are even larger for disentanglement: VHCB achieves substantially higher accuracy and

Table 10: **Concept inference and disentanglement between $c$ and $s$.** StyleGAN2, CelebA-HQ

| Pseudo Label Clf. | Model | Concept Set | Concept Inference | | | Disentanglement | | |
|---|---|---|---|---|---|---|---|---|
| | | | Acc. (↑) | Cosine Sim. (↑) | TV (↓) | Acc. (↑) | Cosine Sim. (↑) | TV (↓) |
| ResNet18 | VHCB | all | 0.855 | **0.804** | **0.148** | **0.927** | **0.917** | **0.076** |
| | CB-AE | all | **0.857** | 0.763 | 0.161 | 0.901 | 0.853 | 0.101 |
| | VHCB | balanced | 0.757 | **0.883** | **0.217** | **0.867** | **0.937** | **0.111** |
| | CB-AE | balanced | **0.761** | 0.872 | 0.234 | 0.776 | 0.811 | 0.192 |
| | VHCB | low corr. | **0.791** | **0.831** | **0.192** | **0.874** | **0.947** | **0.110** |
| | CB-AE | low corr. | 0.779 | 0.739 | 0.238 | 0.701 | 0.803 | 0.229 |
| CLIP | VHCB | all | **0.623** | **0.613** | **0.337** | **0.921** | **0.901** | **0.083** |
| | CB-AE | all | 0.565 | 0.469 | 0.403 | 0.875 | 0.775 | 0.101 |
| | VHCB | balanced | **0.558** | **0.713** | **0.353** | **0.824** | **0.918** | **0.138** |
| | CB-AE | balanced | 0.552 | 0.601 | 0.411 | 0.709 | 0.735 | 0.248 |
| | VHCB | low corr. | 0.599 | **0.730** | **0.299** | **0.854** | **0.943** | **0.125** |
| | CB-AE | low corr. | **0.607** | 0.639 | 0.346 | 0.667 | 0.782 | 0.248 |

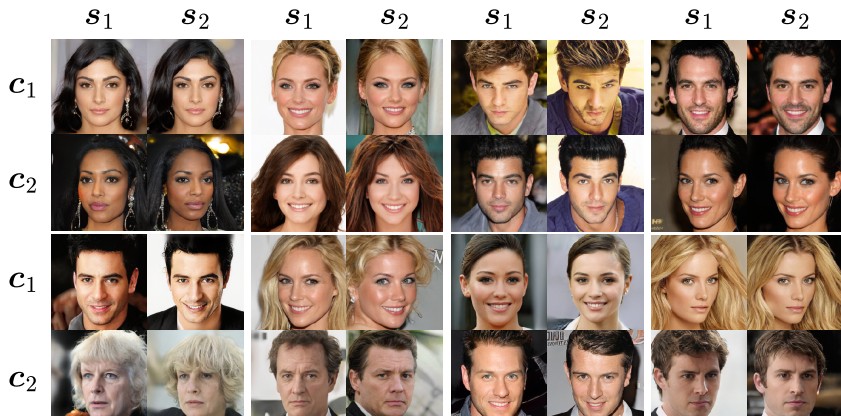

Figure 10: **Qualitative evaluation of the disentanglement between $s$ and $c$** with the VHCB layer and StyleGAN2 models pretrained on CelebA-HQ.

cosine similarity with lower TV, showing that modifying $s$ has minimal effect on concepts. The qualitative results shown in Figure 10 further indicate that modifying $s$ does not induce semantic changes in the generated images. This stronger separation arises from the compact side channel of VHCB (only 5 bits compared to 40 dimensions in CB-AE), which restricts concept leakage into $s$ and enforces a cleaner division between supervised and unsupervised factors.

**Single concept intervention.** We next evaluate the models' ability to manipulate outputs via single-concept interventions, with results reported in Figure 11 and Tables 11 and 12 (intervening on every concept in each set). The VHCB consistently outperforms the CB-AE in terms of target accuracy, learning a robust mapping between latent concepts $c$ and post-bottleneck embeddings $\hat{w}$ without requiring explicit intervention losses. We observe that the CB-AE shows slightly higher non-target accuracy across settings, but this does not indicate more robust interventions. Instead, it reflects a weaker ability to modify the target concept: in many cases, the target is unchanged (lower target accuracy), leaving the output the same and artificially inflating non-target accuracy, which is computed over all 1k interventions. In contrast, the VHCB achieves a better compromise, with large gains in target accuracy (average increase ∼46%) and only a minor reduction in non-target accuracy (average drop ∼7%). For example, on the low-correlation set with ResNet18, target accuracy rises from 0.42 to 0.77, while non-target accuracy decreases slightly from 0.83 to 0.76. Similarly, with CLIP, target accuracy improves from 0.32 to 0.57, with non-target accuracy dropping from 0.83

Table 11: **Single concept interventions** (activation). StyleGAN2, CelebA-HQ.

| Pseudo Label Clf. | Model | Concept Set | Single ($i \to a$) | | | |
|---|---|---|---|---|---|---|
| | | | Target Acc. (↑) | Non Target Acc. (↑) | Non Target Cosine Sim. (↑) | Non Target TV (↓) |
| ResNet18 | VHCB | all | **0.170** | 0.903 | 0.868 | 0.098 |
| | CB-AE | all | 0.105 | **0.924** | **0.910** | **0.078** |
| | VHCB | balanced | **0.376** | 0.812 | 0.869 | 0.160 |
| | CB-AE | balanced | 0.283 | **0.846** | **0.898** | **0.135** |
| | VHCB | low corr. | **0.769** | 0.762 | 0.853 | 0.187 |
| | CB-AE | low corr. | 0.420 | **0.825** | **0.921** | **0.136** |
| CLIP | VHCB | all | **0.131** | 0.890 | 0.830 | 0.112 |
| | CB-AE | all | 0.085 | **0.922** | **0.906** | **0.079** |
| | VHCB | balanced | **0.346** | 0.706 | 0.740 | 0.245 |
| | CB-AE | balanced | 0.206 | **0.839** | **0.884** | **0.135** |
| | VHCB | low corr. | **0.567** | 0.691 | 0.813 | 0.234 |
| | CB-AE | low corr. | 0.317 | **0.829** | **0.921** | **0.130** |

Table 12: **Single concept interventions** (deactivation). StyleGAN2, CelebA-HQ.

| Pseudo Label Clf. | Model | Concept Set | Single ($a \to i$) | | | |
|---|---|---|---|---|---|---|
| | | | Target Acc. (↑) | Non Target Acc. (↑) | Non Target Cosine Sim. (↑) | Non Target TV (↓) |
| ResNet18 | VHCB | all | **0.550** | 0.902 | 0.868 | 0.098 |
| | CB-AE | all | 0.453 | **0.924** | **0.910** | **0.078** |
| | VHCB | balanced | 0.810 | 0.758 | 0.859 | 0.190 |
| | CB-AE | balanced | **0.833** | **0.777** | **0.881** | **0.178** |
| | VHCB | low corr. | **0.765** | 0.781 | 0.867 | 0.177 |
| | CB-AE | low corr. | 0.554 | **0.822** | **0.918** | **0.138** |
| CLIP | VHCB | all | **0.536** | 0.885 | 0.821 | 0.116 |
| | CB-AE | all | 0.311 | **0.922** | **0.906** | **0.080** |
| | VHCB | balanced | **0.682** | 0.627 | 0.732 | 0.296 |
| | CB-AE | balanced | 0.509 | **0.779** | **0.868** | **0.177** |
| | VHCB | low corr. | **0.532** | 0.694 | 0.821 | 0.233 |
| | CB-AE | low corr. | 0.404 | **0.834** | **0.918** | **0.130** |

to 0.69. The same pattern is observed in the complete and balanced sets, showing that the VHCB achieves a better balance between modifying target concepts and preserving non-target concepts.

We also observe that steerability is strongly influenced by concept distribution in the dataset: in CelebA-HQ most concepts are underrepresented, making deactivation easier than activation. The effect of imbalance and model biases is also evident when moving from the full set of 40 unbalanced concepts to balanced and low-correlation subsets, with metrics consistently improving in both models. Therefore, the definition of the concept set has also a large influence in the CBGM performance.

**Hamming distance intervention.** To mitigate the effect of out-of-distribution concept configurations that arbitrary interventions yield during evaluation, we restrict interventions to target one of the 100 most probable concept patterns observed in the dataset, selected via minimum Hamming distance. Results are reported in Table 13. This approach consistently improves intervention success rates in both models, and all the concept sets, with the VHCB still yielding superior metrics, as shown in Table 13. These results further support that steerability depends not only on the expressiveness of the CB layer but also on biases inherent in the pretrained generative model.

**Impact of concept incompleteness and model biases.** With large concept sets, correlations (spurious or not) between concepts can limit steerability, as some configurations fall out of distribution.

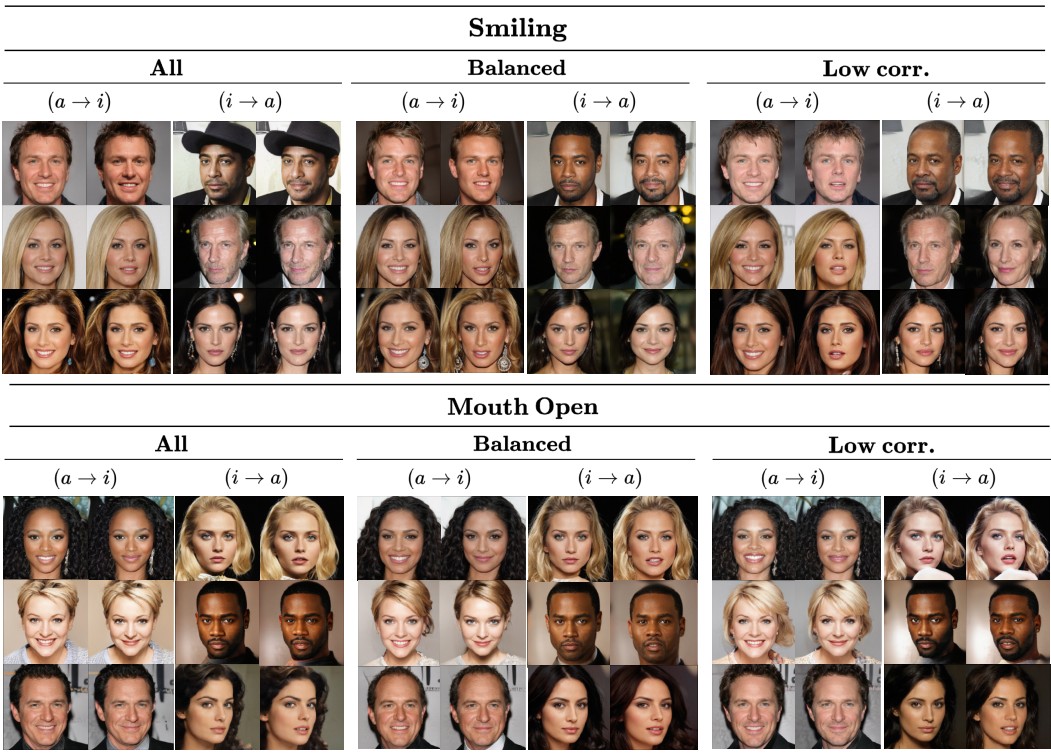

Figure 11: Examples of single concept interventions in CelebA-HQ with VHCB layers trained on different concept sets.

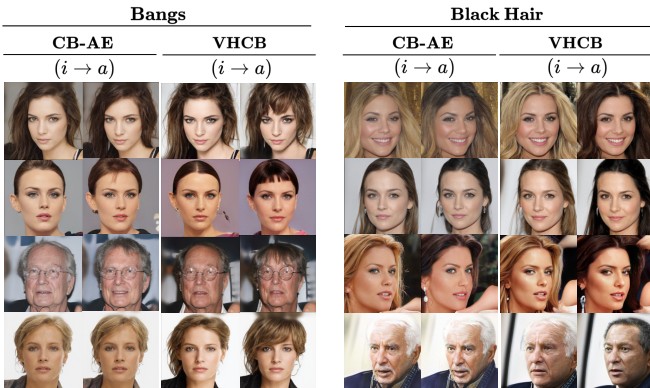

Figure 12: Examples of single concept interventions in CelebA-HQ with VHCB and CB-AE layers.

However, concept information is generally better isolated. In smaller concept sets, low correlations make interventions easier, but the model may inadvertently capture information from unmodeled concepts due to dataset biases. For example, in CelebA-HQ, hair color and 'smiling' exhibit spurious correlations (this effect can be observed in some interventions in Figure 11). Introducing hair color concepts (e.g., 'blond hair', 'black hair') allows to better isolate the concept 'smiling'. However, single concept interventions in this setting become harder, since activating mutually exclusive hair colors yields out-of-distribution configurations. Therefore, the definition of the concept set has also a large influence in the CBGM performance. To further illustrate this effect, Table 13 presents examples of interventions aimed at activating the 'Mustache' concept, which is strongly correlated with 'Male' and 'No Beard'. When modeling the complete set of concepts, most interventions fail, as 'No Beard' (which was already active) and 'Mustache' are mutually exclusive, and due to the

Table 13: **Hamming distance-based interventions.** StyleGAN2, CelebA-HQ.

| Pseudo Label Clf. | Model | Concept Set | Hamming Distance | | | |
| --- | --- | --- | --- | --- | --- | --- |
| | | | Target Acc. (↑) | Non Target Acc. (↑) | Non Target Cosine Sim. (↑) | Non Target TV (↓) |
| ResNet18 | VHCB | all | **0.660** | 0.938 | 0.863 | 0.085 |
| | CB-AE | all | 0.542 | **0.955** | **0.910** | **0.066** |
| | VHCB | balanced | **0.594** | **0.868** | 0.892 | **0.129** |
| | CB-AE | balanced | 0.374 | 0.837 | **0.896** | 0.145 |
| | VHCB | low corr. | **0.634** | 0.845 | 0.895 | 0.149 |
| | CB-AE | low corr. | 0.418 | **0.880** | **0.936** | **0.110** |
| CLIP | VHCB | all | **0.595** | 0.923 | 0.827 | 0.099 |
| | CB-AE | all | 0.466 | **0.949** | **0.902** | **0.070** |
| | VHCB | balanced | **0.423** | 0.719 | 0.753 | 0.239 |
| | CB-AE | balanced | 0.252 | **0.854** | **0.896** | **0.130** |
| | VHCB | low corr. | **0.534** | 0.729 | 0.828 | 0.225 |
| | CB-AE | low corr. | 0.496 | **0.869** | **0.933** | **0.113** |

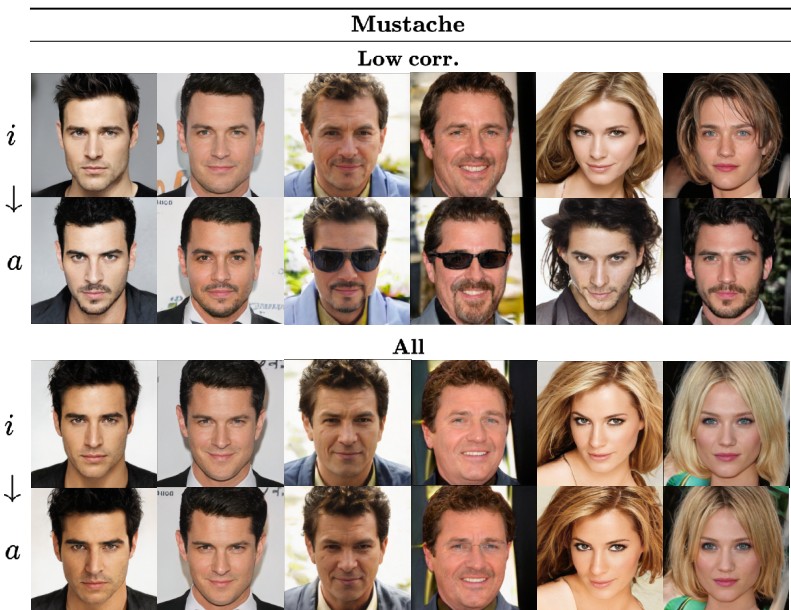

Figure 13: Examples of interventions activating the concept mustache, we can see the effect of the concept incompleteness and the effect of the biases present in the data, captured by the generative model.

data biases, 'Female' and 'Mustache' simultaneously active is also out-of-distribution. However, when using a reduced concept set that excludes 'No Beard' and 'Male', interventions on 'Mustache' succeed. In cases where the mustache is added to a female image, the model changes the gender to 'Male' to accommodate the intervention. Additionally, we observe the model captures spurious correlations associated with the 'Mustache' concept, such as dark hair and a higher likelihood of wearing sunglasses.

**Image generation.** The probabilistic formulation of the VHCB enables direct generation from specified concept configurations. Table 14 reports metrics showing that generated images actually reflect concepts in $c$. Since random sampling of concept configurations can produce out-of-distribution concept configurations, we also sample concept patterns according to their empirical frequency in the training data. This further improves accuracy and highlights how biases in the base generative model can limit the steering capacity of the CBGM. As with interventions, we observe a general improvement in generation accuracy when considering balanced and low-correlation concept sets.

Bangs = 1, MouthOpen = 1, Young = 0, Smiling = 1, Mustache = 0

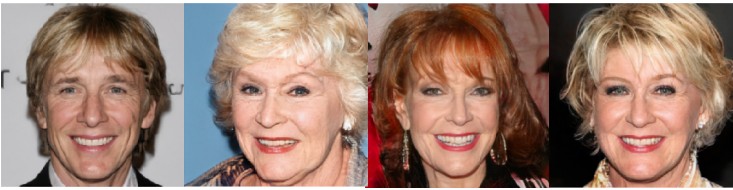

Bangs = 0, MouthOpen = 0, Young = 1, Smiling = 0, Mustache = 0

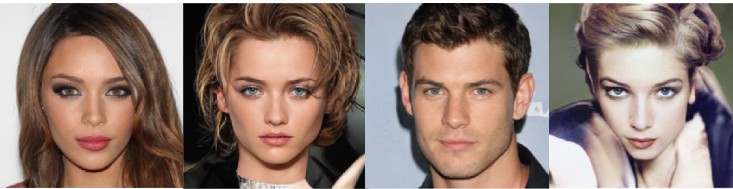

Figure 14: Examples of generation from specific concept configurations in CelebA-HQ. The results were obtained with a pretrained StyleGAN2.

Table 14: **Generation.** StyleGAN2, CelebA-HQ.

| Model | Pseudo Label Clf. | Concept Set | Random Target Acc. (↑) | Patterns Target Acc. (↑) |
|---|---|---|---|---|
| VHCB | ResNet18 | all | 0.551 | 0.873 |
| | | balanced | 0.670 | 0.778 |
| | | low corr. | 0.715 | 0.814 |
| | CLIP | all | 0.533 | 0.830 |
| | | balanced | 0.607 | 0.679 |
| | | low corr. | 0.632 | 0.687 |

Table 15: **Image quality.** We report FID30k using a StyleGAN2 pretrained on CelebA-HQ.

| Pseudo Label Clf. | Model | Concept Set | Forward FID (↓) | Random Generation FID (↓) |
|---|---|---|---|---|
| ResNet18 | VHCB | all | **7.248** | **16.095** |
| | CB-AE | all | 11.645 | – |
| | VHCB | balanced | 11.016 | **19.636** |
| | CB-AE | balanced | **9.169** | – |
| | VHCB | low corr. | **11.589** | **34.925** |
| | CB-AE | low corr. | 13.590 | – |
| CLIP | VHCB | all | **6.388** | **19.601** |
| | CB-AE | all | 12.070 | – |
| | VHCB | balanced | **10.345** | **19.163** |
| | CB-AE | balanced | 11.439 | – |
| | VHCB | low corr. | 9.666 | **24.037** |
| | CB-AE | low corr. | **8.083** | – |

In Table 15, we report FID scores for images generated using the full CBGM (Forward) and by directly sampling from the VHCB latent space (Random Generation). In the forward setting, the VHCB achieves a substantial improvement over the baseline in the complete concept set. However, this gap narrows when considering reduced concept sets, with the baseline outperforming VHCB in two cases. In these experiments, the baseline uses a side channel of dimension 40 (Kulkarni et al., 2025), while our model uses one of dimension 5. The difference in latent dimensionality

Table 16: **Concept inference and disentanglement between $c$ and $s$.** StyleGAN2, CUB-200-2011.

| | | Concept Inference | | | Disentanglement | | |
|---|---|---|---|---|---|---|---|
| Pseudo Label Clf. | Model | Acc. (↑) | Cosine Sim. (↑) | TV (↓) | Acc. (↑) | Cosine Sim. (↑) | TV (↓) |
| ResNet18 | VHCB | **0.743** | 0.914 | **0.167** | 0.864 | 0.957 | 0.100 |
| | CB-AE | 0.720 | **0.919** | 0.181 | **0.962** | **0.991** | **0.027** |
| CLIP | VHCB | 0.554 | **0.812** | **0.246** | **0.792** | **0.943** | **0.127** |
| | CB-AE | **0.570** | 0.543 | 0.355 | 0.612 | 0.867 | 0.216 |

Table 17: **Single concept interventions** (activation). StyleGAN2, CUB-200-2011.

| | | Single $(i \rightarrow a)$ | | | |
|---|---|---|---|---|---|
| Pseudo Label Clf. | Model | Target Acc. (↑) | Non Target Acc. (↑) | Non Target Cosine Sim. (↑) | Non Target TV (↓) |
| ResNet18 | VHCB | **0.392** | **0.754** | **0.896** | **0.164** |
| | CB-AE | 0.112 | 0.669 | 0.895 | 0.184 |
| CLIP | VHCB | **0.372** | 0.665 | 0.859 | 0.196 |
| | CB-AE | 0.226 | **0.905** | **0.905** | **0.157** |

affects model expressivity and, consequently, generative performance. However, we note that the VHCB achieves comparable performance with a much more compact latent space. In contrast, FID increases substantially when generating from random latent samples, as out-of-distribution concept configurations reduce the similarity between the distributions of generated and real images in the learned feature space.

## E.2 STYLEGAN2 ON CUB-200-2011

We also evaluate the different CB layers in the CUB-200-2011 dataset, evaluating the same tasks we did in CelebA-HQ. This is a more challenging dataset, as the annotations are more noisy, as some of the attributes cannot be seen in the pictures, and the size of the dataset is smaller, which does not affect the CB layer itself (as it is trained with generated data) but it does affect the performance of the classifiers employed for training (in the supervised case) and for automatic evaluation.

**Concept Inference**. In this case, we again observe the robustness of the VHCB layer. Metrics in Table 16 show that, while both models perform similarly when trained with supervised classifiers, the VHCB shows a substantially smaller drop in performance when trained with zero-shot classifiers, which tend to be more noisy.

**Interventions.** We next evaluate the models' ability to manipulate model outputs via concept interventions. Tables 17 and 18 contain the metrics for single concept interventions, Table 19 contains the results for Hamming-distance interventions, and Figure 15 provides qualitative examples of the interventions. Although in this case the models perform more closely, we observe that the VHCB has better capability to activate concepts.

**Generation.** Table 20 reports metrics showing that generated images actually reflect concepts in $c$, and Figure 16 shows examples of images generated with fixed concepts. Since random sampling of concept configurations can produce out-of-distribution concept configurations, we also sample concept patterns according to their empirical frequency in the training data. This further improves accuracy and highlights how biases in the base generative model can limit the steering capacity of the CBGM.

Table 18: **Single concept interventions** (deactivation). StyleGAN2, CUB-200-2011.

| Pseudo Label Clf. | Model | Single $(a \rightarrow i)$ | | | |
|---|---|---|---|---|---|
| | | Target Acc. (↑) | Non Target Acc. (↑) | Non Target Cosine Sim. (↑) | Non Target TV (↓) |
| ResNet18 | VHCB | 0.707 | **0.715** | 0.895 | **0.176** |
| | CB-AE | **0.886** | 0.586 | **0.901** | 0.192 |
| CLIP | VHCB | **0.513** | 0.660 | 0.885 | 0.190 |
| | CB-AE | 0.430 | **0.711** | **0.921** | **0.156** |

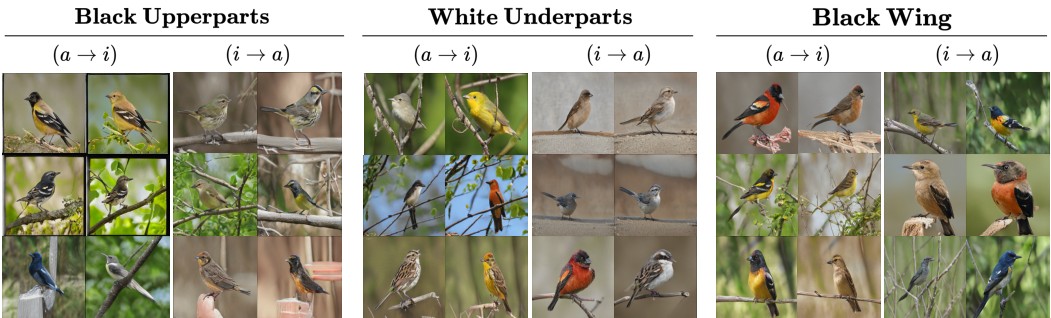

Figure 15: Examples of single concept intervention in CUB-200-2011. Results obtained with a StyleGAN2 pretrained on CUB.

Table 19: **Hamming distance-based interventions.** StyleGAN2, CUB-200-2011.

| Pseudo Label Clf. | Model | Hamming Distance | | | |
|---|---|---|---|---|---|
| | | Target Acc. (↑) | Non Target Acc. (↑) | Non Target Cosine Sim. (↑) | Non Target TV (↓) |
| ResNet18 | VHCB | 0.642 | **0.764** | 0.884 | **0.171** |
| | CB-AE | **0.734** | 0.697 | **0.888** | 0.187 |
| CLIP | VHCB | **0.342** | 0.675 | 0.874 | 0.187 |
| | CB-AE | 0.308 | **0.722** | **0.913** | **0.154** |

$BlackUpperparts = 0, WhiteUnderparts = 1, BlackWing = 0, BlackBill = 0$

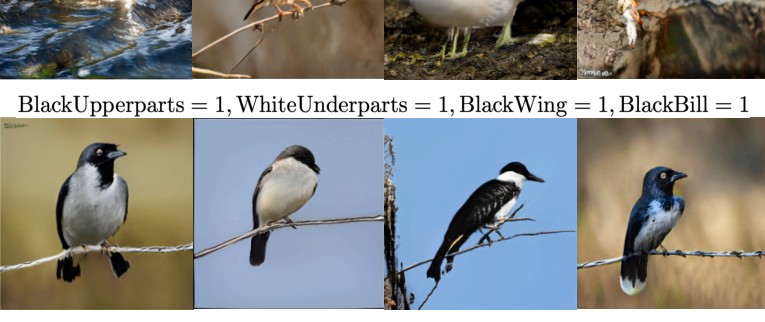

$BlackUpperparts = 1, WhiteUnderparts = 1, BlackWing = 1, BlackBill = 1$

Figure 16: Examples of generation from specific concept configurations in CUB. Results obtained with a StyleGAN2 pretrained on CUB.

Table 20: **Generation.** StyleGAN2 CUB.

| Pseudo Label Clf. | Model | Random Target Acc. (↑) | Patterns Target Acc. (↑) |
|---|---|---|---|
| ResNet18 | VHCB | **0.669** | **0.750** |
|  | CB-AE | – | – |
| CLIP | VHCB | **0.516** | **0.550** |
|  | CB-AE | – | – |

Table 21: **Image quality.** StyleGAN2 CUB.

| Pseudo Label Clf. | Model | Forward FID (↓) | Random Generation FID (↓) |
|---|---|---|---|
| ResNet18 | VHCB | **14.864** | **16.882** |
|  | CB-AE | 27.243 | – |
| CLIP | VHCB | **16.952** | **24.762** |
|  | CB-AE | 24.597 | – |

### E.3 DDPM ON CELEBA-HQ

We next evaluate our approach on a diffusion-based generator, specifically on the pretrained DDPMs described in Section C. We replicate the evaluation protocol from our StyleGAN2 experiments, using the same concept sets and pseudo-label classifiers. Results are reported in Tables 22 and 23 for direct comparison with StyleGAN2. Unlike StyleGAN2, the post-hoc CB-AE baseline from Kulkarni et al. (2025) is not available for this setting, but this does not affect our objective here: our aim is to verify whether the same overall trends observed with StyleGAN2 also hold for DDPMs, rather than to benchmark against all baselines.

**Concept inference.** Table 22 shows that the VHCB captures concept information reliably in the diffusion setting. With ResNet18 supervision, we observe competitive accuracy on the balanced and low-correlation sets, while disentanglement metrics remain consistent across concept sets. As in the StyleGAN2 case, CLIP supervision yields lower overall scores due to noisier labels, but the trends remain stable. Importantly, the side channel retains only limited concept information, confirming that the compact binary bottleneck continues to mitigate leakage in DDPMs. Injecting the VHCB model at all levels of the U-Net's upsampling path provides only slight improvements in these metrics compared to the case where the VHCB module is only injected at the bottleneck of the U-Net architecture.

**Concept interventions.** The steerability of the VHCB within DDPMs is summarized in Table 23. We evaluate activation ($i \rightarrow a$), deactivation ($a \rightarrow i$), and interventions restricted to training concept patterns (minimum Hamming distance). As in the StyleGAN2 setting, we observe substantial improvements in target accuracy when deactivating concepts, with non-target attributes largely preserved. However, unlike in StyleGAN2, Hamming distance interventions do not improve target accuracy: while they achieve excellent preservation of non-target concepts (above 0.98 across all settings), they fail to reliably enforce the target concept. This contrast suggests that enforcing concept changes in DDPMs is harder than in GANs, likely because the denoising trajectory resists out-of-distribution edits. Moreover, this may reflect a limitation of introducing the VHCB only at the bottleneck of the U-Net backbone. Injecting the output of the VHCB module at all levels of the U-Net architecture's upsampling path is a first step to sidestep this issue. Indeed, quantitative metrics across concepts generally improve in this case. However, we believe that further research must be performed properly characterize the architectural trade-offs, identify the most effective injection strategy, and ensure stable and consistent improvements.

Table 22: **Concept inference and disentanglement between $c$ and $s$.**

DDPM, CelebA-HQ.

| Model | Pseudo Label Clf. | Concept Set | Concept Inference | | | Disentanglement | | |
|---|---|---|---|---|---|---|---|---|
| | | | Acc. ($\uparrow$) | Cosine Sim. ($\uparrow$) | TV ($\downarrow$) | Acc. ($\uparrow$) | Cosine Sim. ($\uparrow$) | TV ($\downarrow$) |
| VHCB *(bottleneck)* | ResNet18 | all | **0.733** | 0.595 | **0.254** | **0.733** | 0.595 | **0.255** |
| | | balanced | 0.546 | **0.627** | 0.402 | 0.545 | **0.633** | 0.402 |
| | | low corr. | 0.621 | 0.333 | 0.383 | 0.622 | 0.338 | 0.383 |
| | CLIP | all | **0.684** | 0.431 | **0.320** | **0.680** | 0.431 | **0.321** |
| | | balanced | 0.577 | 0.224 | 0.430 | 0.569 | 0.231 | 0.434 |
| | | low corr. | 0.339 | **0.481** | 0.569 | 0.345 | **0.483** | 0.567 |
| VHCB *(all levels)* | ResNet18 | all | **0.753** | 0.547 | 0.242 | **0.750** | 0.548 | 0.243 |
| | | balanced | 0.567 | **0.601** | 0.404 | 0.568 | **0.600** | 0.405 |
| | | low corr. | 0.563 | 0.480 | **0.410** | 0.557 | 0.482 | **0.412** |
| | CLIP | all | **0.688** | 0.437 | 0.308 | **0.688** | 0.437 | 0.308 |
| | | balanced | 0.545 | 0.667 | 0.393 | 0.552 | **0.672** | **0.391** |
| | | low corr. | 0.547 | **0.668** | 0.393 | 0.552 | **0.672** | **0.391** |

Table 23: **Test-time interventions.** Evaluation of single-concept activation $(i \rightarrow a)$, deactivation $(a \rightarrow i)$, and interventions guided by training concept patterns (minimum Hamming distance). DDPM, CelebA-HQ.

| Model | Pseudo Label Clf. | Concept Set | Single $(i \rightarrow a)$ | | Single $(a \rightarrow i)$ | | Hamming Dist. | |
|---|---|---|---|---|---|---|---|---|
| | | | Target Acc. ($\uparrow$) | Non Target Acc. ($\uparrow$) | Target Acc. ($\uparrow$) | Non Target Acc. ($\uparrow$) | Target Acc. ($\uparrow$) | Non Target Acc. ($\uparrow$) |
| VHCB *(bottleneck)* | ResNet18 | all | 0.131 | 0.905 | 0.546 | **0.898** | **0.047** | **0.993** |
| | | balanced | **0.406** | 0.625 | 0.250 | 0.491 | 0.044 | 0.983 |
| | | low corr. | 0.187 | **0.910** | 0.571 | 0.857 | 0.041 | 0.990 |
| | CLIP | all | 0.131 | 0.905 | 0.546 | **0.898** | 0.041 | 0.992 |
| | | balanced | **0.343** | 0.607 | 0.250 | 0.500 | 0.037 | 0.980 |
| | | low corr. | 0.219 | **0.906** | 0.571 | 0.847 | 0.046 | 0.991 |
| VHCB *(all levels)* | ResNet18 | all | 0.150 | **0.904** | 0.568 | 0.895 | 0.332 | **0.952** |
| | | balanced | **0.562** | 0.598 | 0.313 | 0.455 | **0.378** | 0.851 |
| | | low corr. | 0.531 | 0.584 | 0.464 | 0.469 | 0.205 | 0.877 |
| | CLIP | all | 0.118 | **0.901** | 0.545 | 0.899 | 0.296 | **0.953** |
| | | balanced | 0.281 | 0.651 | 0.250 | 0.500 | 0.197 | 0.842 |
| | | low corr. | **0.437** | 0.616 | 0.357 | 0.489 | 0.213 | 0.844 |

**Qualitative Results.** Examples of test-time interventions on the CelebA-HQ dataset are presented in the following figures. Each example consists of a triplet of images displayed side by side using the model configuration of concept code rate $40/800$ and an unsupervised binary latent with code rate $5/50$. The left image shows the output of the baseline DDPM model without any intervention or VHCB layer. The center image illustrates the result of applying the VHCB intervention at the bottleneck layer only (as depicted in Figure 8). The right image shows the effect of injecting the VHCB at all levels of the upsampling path of the U-Net (Figure 9). We demonstrate interventions on several semantic concepts: Smiling (Figure 17), Male (Figure 18), and Mouth Slightly Open (Figure 19) (each from the inactive to the active state). These images were taken following the same evaluation scheme: intervening on concepts starting only at timestep $t = 400$ and using the denoising diffusion implicit model (DDIM) Scheduler for 50 total number of timesteps. Comparing both DDPM variants with the VHCB, there is no significant impact for the VHCB when injected in all levels when compared to the injected at bottleneck only model. Note that the quality of the images depends on the pretrained model as previously stated, where the training data used were pre-generated images from the original model.

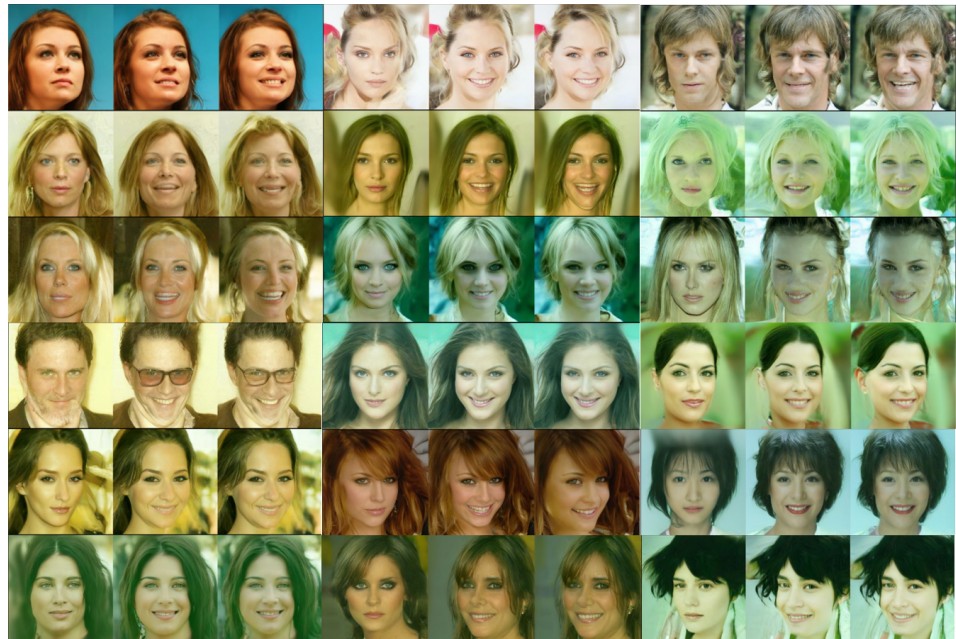

Figure 17: Examples of single concept intervention in CelebA-HQ dataset for the *smiling* attribute, from inactive to active. Results obtained with a DDPM pretrained on CelebA-HQ. VHCB model configuration: concept code rate of $R = 40/800$ and unsupervised latent code rate of $R = 5/50$. Each image triplet shows: baseline DDPM (left), VHCB at bottleneck (center), and VHCB injected across all upsampling layers (right).

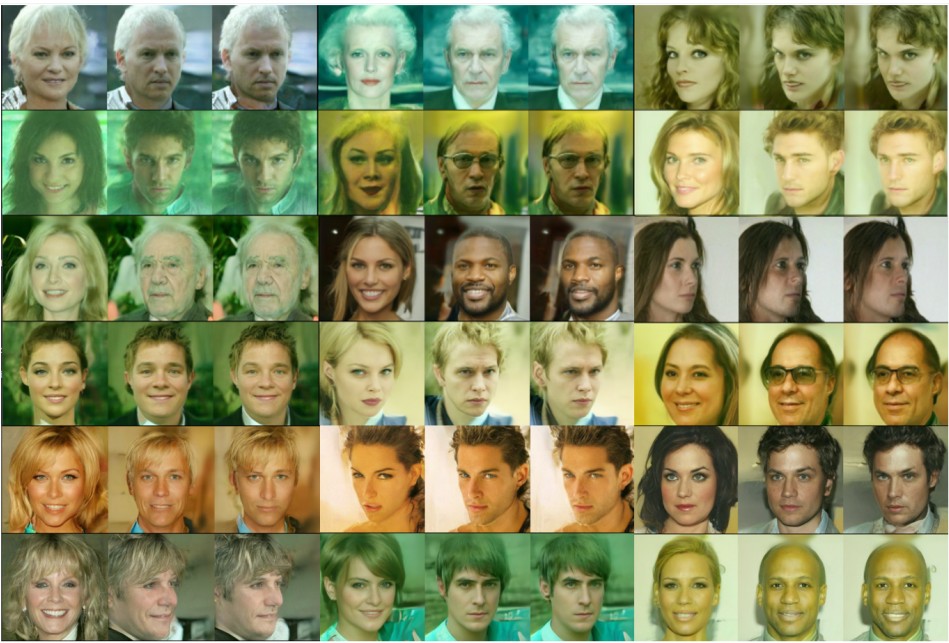

Figure 18: Examples of single concept intervention in CelebA-HQ dataset for the *male* attribute, from inactive to active. Results obtained with a DDPM pretrained on CelebA-HQ. VHCB model configuration: concept code rate of $R = 40/800$ and unsupervised latent code rate of $R = 5/50$. Each image triplet shows: baseline DDPM (left), VHCB at bottleneck (center), and VHCB injected across all upsampling layers (right).

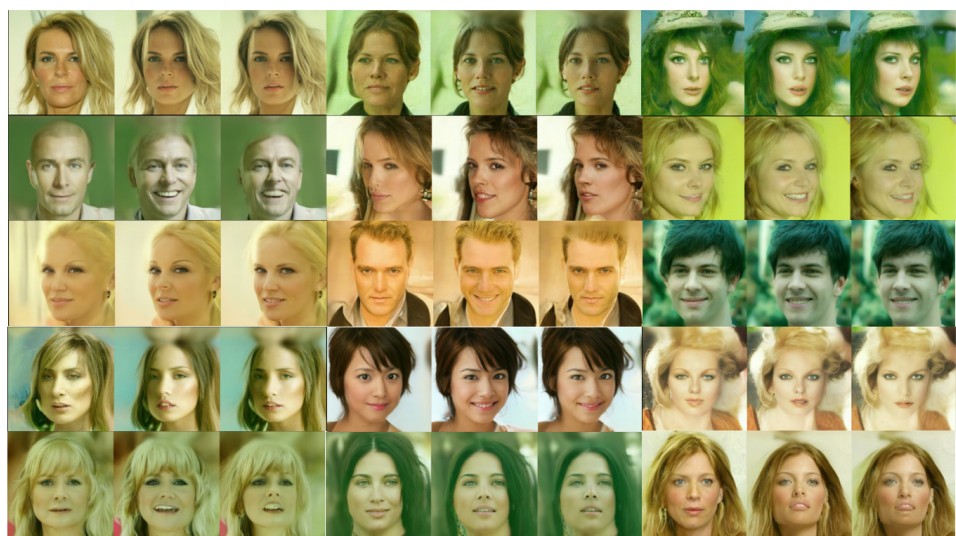

Figure 19: Examples of single concept intervention in CelebA-HQ dataset for the *mouth slightly open* attribute, from inactive to active. Results obtained with a DDPM pretrained on CelebA-HQ. VHCB model configuration: concept code rate of $R = 40/800$ and unsupervised latent code rate of $R = 5/50$. Each image triplet shows: baseline DDPM (left), VHCB at bottleneck (center), and VHCB injected across all upsampling layers (right).

Table 24: **Generation.** DDPM, CelebA-HQ.

| Model | Pseudo Label Clf. | Concept Set | Random Target Acc. (↑) | Patterns Target Acc. (↑) |
|---|---|---|---|---|
| VHCB *(bottleneck)* | ResNet18 | all | **0.506** | **0.711** |
| | | balanced. | 0.490 | 0.490 |
| | | low corr. | 0.503 | 0.580 |
| | CLIP | all | **0.505** | **0.705** |
| | | balanced. | 0.491 | 0.545 |
| | | low corr. | 0.502 | 0.530 |
| VHCB *(all levels)* | ResNet18 | all | 0.506 | **0.752** |
| | | balanced. | 0.498 | 0.561 |
| | | low corr. | **0.511** | 0.533 |
| | CLIP | all | 0.503 | **0.761** |
| | | balanced. | 0.499 | 0.563 |
| | | low corr. | **0.510** | 0.552 |

**Image generation.** Results are summarized in Table 24. As with interventions, generated images generally reflect the intended concepts, confirming that the VHCB enables controllable generation within DDPMs. Sampling concept patterns according to their empirical frequency in the training data typically improves target accuracy compared to purely random sampling, particularly for the full concept set. For the balanced and low-correlation subsets, improvements are smaller and occasionally absent.

We attribute this variability not to limitations of the VHCB itself, but to how effectively concept information can be propagated through the U-Net denoiser over the many iterative steps of the diffusion process. As discussed earlier, injecting the VHCB at all levels of the upsampling path provides a stronger and more distributed conditioning mechanism, which is particularly relevant given the multi-step nature of DDPMs. The generation results reflect this: the multilevel variant yields slightly higher target accuracy across all concept subsets, although the improvements remain modest overall.

Taken together, these findings indicate that while the proposed framework enables effective concept-based control, achieving stronger and more consistent gains will likely require a deeper exploration of how concept representations are integrated and preserved throughout the denoising trajectory in diffusion architectures.

