# OpenReview forum: "A Probabilistic Hard Concept Bottleneck for Steerable Generative Models"
_ICLR.cc/2026/Conference — ICLR 2026 Poster_

### Official Review · Reviewer_Te4N · 2025-10-31

**Soundness:** 2
**Presentation:** 3
**Contribution:** 3
**Rating:** 4
**Confidence:** 4

**Summary:**

This paper focuses on concept bottleneck generative models (CBGM) which are interpretable and steerable generative models. Existing CBGMs are deterministic models which limit steering to concepts predicted in existing inputs and may suffer from concept leakage. To overcome these limitations, this paper proposes a Variational Hard Concept Bottleneck (VHCB) layer which allows direct generation from sampled concepts and could reduce concept leakage. They evaluate the proposed method compared to the previous SOTA method and show improvements in steerability.

**Strengths:**

* The paper is fairly well-written and easy to follow with good illustrations.

* The paper focuses on the relatively under-explored topic of concept bottleneck generative models.

* The proposed VHCB layer seems to be novel and it makes intuitive sense to move from deterministic CB layers (like in CB-AE or CBGM) to probabilistic ones like VHCB, especially given that the generative task itself is not deterministic.

* I also appreciate the extensive experiments, especially the disentanglement experiments which were not reported in prior work.

**Weaknesses:**

* It is unclear to me how the error correcting codes help mitigate concept leakage. Rather it seems that increasing the dimensionality of the concepts (and the unsupervised embedding) would increase the chances of concept leakage. It would be good to elaborate on why (and which part of VHCB) actually helps to mitigate concept leakage.
    * Another related concern is that concept leakage might actually stem from the unsupervised embedding itself. It would be good to discuss this in the paper and comment on whether this can be removed from concept bottleneck generative models.

* Based on Table 10, it seems VHCB sometimes achieves better and sometimes worse FID scores than CB-AE, but there is no explanation/intuition offered for this.
    * Related to this, the FID scores are significantly higher when generating from random concepts which is attributed to some concept combinations potentially being out of distribution. I wonder if this can be improved by generating images from random concept samplings during training and applying a loss on the concept predictions (or an image generation loss to improve the quality of those images) similar to the intervention losses in CB-AE.
    * The higher FID scores when generating from random concepts also seems to be a major limitation because that was proposed as one of the main contributions of this work (which was not possible in the deterministic CBGM or CB-AE).

* Minor issues (typos):
    * L113: "Blottleneck"

**Questions:**

Please address my questions/comments from the weaknesses section.

---

> ### Author Response · Authors · 2025-11-21
>
> **ECCs improve concept accuracy**
>
> Following prior CBM work (Margeloiu et al., 2021; Havasi et al., 2022; Lockhart et al., 2022; Vandenhirtz et al., 2024; Sun et al., 2024), we use hard concept representations to prevent concept leakage and enhance steerability, as confirmed by our experiments (Tables 3, 6, 7, 8, 12; Figures 4, 8, 9, 12). This is implemented via a binary VAE (specifically, the Coded DVAE) which propagates hard binary samples rather than concept probabilities in its generative model. Incorporating ECCs improves concept accuracy in the VAE latent space, leading to a tighter variational posterior and more effective training, as shown by Martínez-García et al. (2025). As a result, a robust binary VAE forms the core of the VHCB layer: binary representations naturally mitigate concept leakage, while ECCs enhance concept accuracy.
>
> **Remove the side channel or unsupervised embedding**
>
> The unsupervised embedding, or side channel, is used to handle concept incompleteness. As discussed in Section 2, generation depends on factors that cannot be fully captured by human-interpretable concepts (e.g., texture, lighting, or object position), making the predefined concept set inherently incomplete. Removing the side channel produces the following effects:
>
> - For *large concept sets*, we observe that removing the side channel does not have a significant impact in performance, following the results obtained in the ablation of the weight of the side channel regularization reported in our response to reviewer 5vRH.
>
> - For *reduced concept sets*, we observe a significant drop in generation quality, with FID nearly doubling, due to the limited expressiveness of the latent space. Removing the side channel significantly reduces the generative model’s expressivity and, consequently, also (slightly) its steerability.
>
> |  Pseudo-label clf. | Concept set | Side Channel size | Acc intervention (i to a) | Acc intervention (a to i) | Hamming Distance Intervention| Acc Generation | FID Forward |
> |-----|-----|-----|-----|-----|-----|-----|-----|
> | ResNet18 | all | 5 | 0.170 | 0.550 | 0.660 | 0.551 | 7.248 |
> | ResNet18 | all | 0 | 0.191 | 0.538 | 0.664 | 0.539 | 7.921 |
> |-----|-----|-----|-----|-----|-----|-----|-----|
> | ResNet18 | balanced | 5 | 0.375 | 0.810 | 0.594 | 0.670 | 11.016 |
> | ResNet18 | balanced | 0 | 0.329 | 0.711 | 0.420 | 0.625 | 20.950 |
> |-----|-----|-----|-----|-----|-----|-----|-----|
> | ResNet18 | low corr. | 5 | 0.769 | 0.765 | 0.634 | 0.715 | 11.589 |
> | ResNet18 | low corr. | 0 | 0.604 | 0.837 | 0.765 | 0.688 | 19.501 |
>
> **FID results with respect to the baseline**
>
> In the FID results reported in the Appendix, VHCB achieves a substantial improvement over the baseline in the complete concept set. However, this gap narrows when considering reduced concept sets, with the baseline outperforming VHCB in two cases. In these experiments, the baseline uses a side channel of dimension 40 (Kulkarni et al., 2025), while our model uses one of dimension 5. The difference in latent dimensionality affects model expressivity and, consequently, generative performance. We will further clarify this in the paper.
>
> In the side-channel regularization ablation reported in our response to Reviewer 5vRH, we include results with a model trained on the balanced set using a side channel of dimension 10, achieving a FID of 8.661 ($\beta=1$), which surpasses the baseline FID of 9.169 for the same concept set, while using a binary side channel with ¼ of dimensions. This supports the claim that VHCB can achieve competitive or superior generative quality with a much more compact latent space.
>
> **FID in random generation**
>
> We do not view the higher FID scores observed when sampling concepts at random as a limitation of our method. Since FID measures the similarity between the generated and training data distributions, randomly sampling concepts independently can create out-of-distribution concept sets, and thus generated images, leading to higher FID values.
>
> **Typos**
>
> We have corrected the typo in the revised version of the paper.

---

> > ### Author Response · Authors · 2025-11-21
> >
> > **REFERENCES**
> >
> > Andrei Margeloiu, Matthew Ashman, Umang Bhatt, Yanzhi Chen, Mateja Jamnik, and Adrian
> > Weller. Do Concept Bottleneck Models Learn as Intended? arXiv preprint arXiv:2105.04289,
> > 2021.
> >
> > Marton Havasi, Sonali Parbhoo, and Finale Doshi-Velez. Addressing Leakage in Concept Bottleneck
> > Models. Advances in Neural Information Processing Systems, 35:23386–23397, 2022.
> >
> > Joshua Lockhart, Nicolas Marchesotti, Daniele Magazzeni, and Manuela Veloso. Towards learn-
> > ing to explain with concept bottleneck models: mitigating information leakage. arXiv preprint
> > arXiv:2211.03656, 2022.
> >
> > Moritz Vandenhirtz, Sonia Laguna, Ričards Marcinkevičs, and Julia Vogt. Stochastic Concept Bottleneck Models. Advances in Neural Information Processing Systems, 37:51787–51810, 2024.
> >
> > Ao Sun, Yuanyuan Yuan, Pingchuan Ma, and Shuai Wang. Eliminating Information Leakage in
> > Hard Concept Bottleneck Models with Supervised, Hierarchical Concept Learning. arXiv preprint
> > arXiv:2402.05945, 2024.
> >
> > María Martínez-García, Grace Villacrés, David Mitchell, and Pablo M. Olmos. Improved Variational
> > Inference in discrete VAEs using Error Correcting Codes. In Proceedings of the Forty-first Con-
> > ference on Uncertainty in Artificial Intelligence, volume 286 of Proceedings of Machine Learning
> > Research, pp. 2973–3012. PMLR, 21–25 Jul 2025.
> >
> > Akshay Kulkarni, Ge Yan, Chung-En Sun, Tuomas Oikarinen, and Tsui-Wei Weng. Interpretable
> > Generative Models through Post-hoc Concept Bottlenecks. In IEEE/CVF Conference on Computer Vision and Pattern Recognition, 2025.

---

> > > ### Author Response · Authors · 2025-11-21
> > >
> > > *We would like to thank the reviewer for their positive and constructive feedback. We have carefully considered each comment, provided detailed responses above, and updated the paper accordingly. The submission has been revised to incorporate all of the reviewers’ suggestions. If the reviewer finds that their concerns have been adequately addressed, we respectfully invite them to update their score.*

---

### Official Review · Reviewer_5vRH · 2025-10-31

**Soundness:** 3
**Presentation:** 3
**Contribution:** 3
**Rating:** 6
**Confidence:** 4

**Summary:**

The paper introduces the Variational Hard Concept Bottleneck (VHCB) layer, a probabilistic extension of Concept Bottleneck Generative Models (CBGMs). Instead of previous approaches, it uses hard binary concept representations (binary latent variables with probabilistic smoothing) to reduce concept leakage and a probabilistic latent formulation to enable direct sampling from concept configurations. The approach is validated on StyleGAN2 and DDPM trained on CelebA-HQ and CUB-200-2011 datasets.

**Strengths:**

- Addresses an important and previously challenging problem in the concept bottleneck literature.
- The side channel regularization is quite interesting. This 'independence regularization' term is quite different from those in the literature. It acts as a capacity control and seeks to prevent concept leakage into the unsupervised channel.
- The evaluation, in the work, compared to the baselines, is quite thorough.
- The paper also has a nice trick for reducing potentially incoherent steering on concepts. They restrict interventions to target one of the 100 most probable concept patterns observed in the dataset, selected via minimum Hamming distance. This ensures that interventions are more coherent. Of course, this is also limits the ability of the model to extrapolate.

**Weaknesses:**

- The probabilistic formulation is based on existing binary VAEs (from Martínez-García et al., 2025). This works applies it to CBGMs directly.
- The paper introduces a binary side channel regularized toward a factorized prior via a KL penalty to mitigate concept incompleteness. This mechanism does not guarantee independence from the concept latents; it only limits capacity. The authors could strengthen their claims by including ablations on the KL weight, measuring mutual information between, and or visualizing variations induced by the unsupervised channel to confirm it captures residual, non-semantic factors.
- The current disentanglement evaluation, in the paper, is also not quite appropriate. It involves: sampling two independent latent pairs, swapping the unsupervised latents  to form mixed pairs, and regenerating images and comparing whether the concepts remain unchanged. This test examines whether observable concept labels change, but not whether these latents themselves are statistically independent. You could have apparent disentanglement (concepts unchanged) while they still encodes correlated concept information in latent space. If those classifiers are not sensitive to subtle variations, you can get artificially high “disentanglement” scores. I would recommend the authors consider other concrete measures of disentanglement. For example, the Glancenets paper that the authors cite has a measure, DCI, and toy experiments that they can use to stress test their approach.

**Questions:**

- Why a symmetric KL instead of standard cross-entropy or forward KL?
- It is hard to disentangle whether leakage prevention comes from binary concepts or the KL regularization. Can the authors speak more to this? Can you train a simplified version of your model in a toy setting like in the work of GlancNets so that you can tease these issues apart?
- I would suggest that the authors also look into measure of disentanglement or independence between latent factors to concretize their claims.

---

> ### Author Response · Authors · 2025-11-21
>
> **Novelty of the approach**
>
> We acknowledge that the building block, the binary VAE, is not novel. However, its extension  and application to CBGMs is indeed novel and requires several modifications of the original model, such  jointly model supervised and unsupervised representations, introducing supervision through the prior for one of the latent variables. As we state in our paper our main contribution is threefold: i) the first approach fo CBGMs with binary concepts (and binary side channel), with ii) a novel probabilistic formulation that enables steerable generation through both direct concept-based sampling and concept interventions; and iii) an evaluation framework  for systematic and, compared to prior literature, more comprehensive evaluation framework for the novel family of CBGMs (the two closest papers are from ICLR’24 and CVPR’25). We believe that these contributions are relevant for the ICLR community and open several lines of future work.
>
> **Justification for use of symmetric KL as concept loss**
>
> To empirically validate the choice of the symmetric KL divergence, we have now conducted an ablation study comparing the models trained with BCE, forward KL, and symmetric KL as concept loss. Our results complements our provided intuition in lines 226-227 of the submitted paper, by illustrating that  models trained with BCE tend to perform slightly better when deactivating concepts, models trained with forward KL generally achieve slightly better control when activating concepts, and models trained with symmetric KL provide the best balance, offering strong interventional performance in both directions. We have now added these results to the Appendix, and further clarified this point in the above argument in the revised main.
>
> | Concept loss | Pseudo-label clf. | Concept set | Concept Acc. | Acc. intervention (i to a) | Acc. intervention (a to i) | Acc. Hamming dist. intervention | Acc Generation (random)|
> |-----|-----|-----|-----|-----|-----|-----|-----|
> | BCE | ResNet18 | all | 0.855 | 0.165 | 0.555 | 0.676 | 0.543 |
> | KL | ResNet18 | all | 0.852 | 0.168 | 0.524 | 0.669 | 0.546 |
> | Sym. KL | ResNet18 | all | 0.855 | 0.170 | 0.550 | 0.660 | 0.551 |
> |-----|-----|-----|-----|-----|-----|-----|-----|
> | BCE | CLIP | all | 0.633 | 0.127 | 0.564 | 0.621 | 0.527 |
> | KL | CLIP | all | 0.623 | 0.128 | 0.508 | 0.603 | 0.530 |
> | Sym. KL | CLIP | all | 0.623 | 0.131 | 0.536 | 0.594 | 0.533|
> |-----|-----|-----|-----|-----|-----|-----|-----|
> | BCE | ResNet18 | balanced | 0.609 | 0.395 | 0.564 | 0.527 | 0.632|
> | KL | ResNet18 | balanced | 0.751 | 0.314 | 0.748 | 0.726 | 0.663 |
> | Sym. KL | ResNet18 | balanced | 0.757 | 0.375 | 0.814 | 0.594 | 0.670 |
> |-----|-----|-----|-----|-----|-----|-----|-----|
> | BCE | CLIP | balanced | 0.521 | 0.405 | 0.654 | 0.721 | 0.613 |
> | KL | CLIP | balanced | 0.549 | 0.306 | 0.745 | 0.767 | 0.602 |
> | Sym. KL | CLIP | balanced | 0.558 | 0.346 | 0.682 | 0.423 | 0.607 |
> |-----|-----|-----|-----|-----|-----|-----|-----|-----|
>
> **Visualizing variations induced by the unsupervised channel to confirm it captures residual factors, not the (supervised) concepts**
>
> In line with the reviewer’s suggestion, we provide qualitative results for the disentanglement experiment to further support our claims. Here, we draw two independent latent pairs and swap their side-channel vectors, and regenerate the images to verify that the concepts remain unchanged. We display the images generated from the original pairs $[\boldsymbol{c}_1,\boldsymbol{s}_1]$ and $[\boldsymbol{c}_2, \boldsymbol{s}_2]$, as well as from the swapped pairs $[\boldsymbol{c}_1,\boldsymbol{s}_2]$ and $[\boldsymbol{c}_2,\boldsymbol{s}_1]$. The results show that the concepts are preserved, while attributes such as clothing, background, and face orientation vary with the side channel. Moreover, the same side-channel vector induces similar features across images with different concepts. In this case, we are modelling the 40 concepts in CelebA-HQ. We have now added these results in the revised version of the submitted paper (both in the main and the Appendix).

---

> > ### Author Response · Authors · 2025-11-21
> >
> > **Ablation on the unsupervised KL**
> >
> > We agree with the reviewer that such an ablation will indeed strengthen our message and thus have included it in Appendix. Our new empirical results vary the weight of the unsupervised KL for different side channel sizes and concept sets. Our results, included below, show that this parameter does not have any significant effect in any of the reported metrics when the concept set is large (i.e., 40 concepts in CelebA-HQ). When considering small concept sets, both steerability and FID slightly deteriorate as side-channel regularization increases. This can be explained by the fact that the proposed CBGM is a generative model optimized to minimize reconstruction loss. When only a small subset of concepts is used and the side channel is over-regularized the expressivity of the model is constrained, which in turn limits both the quality of generated samples (measured by FID) and, consequently, the steerability.
> >
> > | Weight unsupervised KL| Pseudo-label clf. |Side channel size|  Concept set | Concept Acc. | Acc intervention (i to a) | Acc intervention (a to i) | Acc Hamming distance intervention | Acc Generation (random) | FID forward |
> > |-----|-----|-----|-----|-----|-----|-----|-----|-----|-----|
> > |0| ResNet18 | all | 5 | 0.853 | 0.173 | 0.558 | 0.666 |0.5484 | 6.855 |
> > |1| ResNet18 | all | 5 | 0.855 | 0.170 | 0.550 | 0.660 | 0.551 | 7.248 |
> > |10| ResNet18 | all | 5 | 0.856 | 0.173 | 0.562 | 0.656 | 0.550 | 7.546 |
> > |20| ResNet18 | all | 5 | 0.853 | 0.173 | 0.544 | 0.673 | 0.549 | 7.590 |
> > |40| ResNet18 | all | 5 | 0.850 | 0.174 | 0.550 | 0.659 | 0.543 | 7.743 |
> > |-----|-----|-----|-----|-----|-----|-----|-----|-----|
> > |0| ResNet18 | balanced | 5 | 0.773 | 0.358 | 0.784 | 0.627 | 0.659 | 11.1303|
> > |1| ResNet18 | balanced | 5 | 0.752 | 0.375 | 0.814 | 0.594 | 0.670 | 11.019 |
> > |10| ResNet18 | balanced | 5 | 0.754 | 0.362 | 0.822 | 0.609 |  0.609 | 11.870 |
> > |20| ResNet18 | balanced | 5 | 0.763 | 0.364 | 0.817 | 0.596 | 0.596 | 12.374 |
> > |40| ResNet18 | balanced | 5 | 0.763 | 0.337 | 0.799 | 0.561 | 0.561 | 12.687 |
> > |-----|-----|-----|-----|-----|-----|-----|-----|-----|
> > |0| ResNet18 | balanced | 10 | 0.766 | 0.363 | 0.785 | 0.606 | 0.669 | 10.946 |
> > |1| ResNet18 | balanced | 10 | 0.770 | 0.356 | 0.788 | 0.514 | 0.773 | 8.661 |
> > |10| ResNet18 | balanced |10 | 0.774 | 0.351 | 0.822 | 0.509 | 0.663 | 10.309 |
> > |20| ResNet18 | balanced | 10 | 0.774 | 0.350 | 0.818 | 0.499 | 0.664 | 10.481 |
> > |40| ResNet18 | balanced | 10 | 0.774 | 0.326 | 0.811 | 0.500 | 0.665 | 9.216 |
> >
> > In the Appendix of the revised submission, we provide a more extensive experimental evaluation, including disentanglement metrics. As the conclusions align with those obtained from the steerability results shown in the previous table, we have omitted these details here to maintain a concise discussion. We refer the reviewer to the revised Appendix for the full set of experimental results.

---

> ### Author Response · Authors · 2025-11-21
>
> **It is hard to disentangle whether leakage prevention comes from binary concepts or the KL regularization. Can the authors speak more to this? Can you train a simplified version of your model in a toy setting like in the work of GlanceNets so that you can tease these issues apart?**
>
> We have conducted a thorough evaluation on complex datasets that allow us to assess how hard concepts and the side channel impact the steerability of the model in real scenarios.
>
> - **Leakage prevention.** Concept leakage occurs when concept probabilities unintentionally encode task-related information, in this case, additional generative factors. This degrades steerability because interventions on a concept no longer produce the intended effect. For this reason, we use steerability metrics as a proxy for assessing leakage. Prior work on CBMs shows that hard concepts mitigate leakage due to their limited expressivity (Margeloiu et al., 2021; Havasi et al., 2022; Lockhart et al., 2022; Vandenhirtz et al., 2024; Sun et al., 2024). Our steerability experiments support this: we improve the success rates of concept interventions relative to a baseline that uses soft concepts. We provide both quantitative (Tables 3, 6, 7, 8, 12) and qualitative evidence (Figures 4, 8, 9, 12) demonstrating an improved steerability.
>
> - **Side channel regularization through the unsupervised KL.** The side channel accounts for concept incompleteness (as we clarify in lines 34-39 of the submitted paper), capturing generative factors not explained by the predefined concept set that are necessary for generation. During training, its capacity is regularized to ensure the model relies on concepts during generation. The ablation study on the weight of the unsupervised KL provided above shows that:
>
>   - For *large concept sets*, it does not have any significant effect on steerability and generation.
>
>   - For *reduced concept sets*, both steerability and FID degrade slightly as regularization increases, since an over-regularized side channel restricts the generative model’s expressivity and, consequently, also its steerability.
>
> We note that the effect of side-channel regularization on steerability is minimal compared to the impact of soft vs. hard concepts, which remains the dominant factor. This supports the use of steerability metrics as a proxy for measuring leakage.
>
> **I would suggest that the authors also look into measures of disentanglement or independence between latent factors to concretize their claims.**
>
> We agree with the reviewer that including additional disentanglement metrics and experiments would strengthen our empirical evaluation. Accordingly, we conducted two additional experiments to evaluate disentanglement in the latent space: one based on accuracy in inference and the other using the DCI metric suggested by the reviewer.
>
> - **Feedback loop experiment**. To address the reviewer’s concern that our results might overestimate disentanglement due to the sensitivity of the final classifiers, we replicated the experiment using the autoencoder architecture of the VHCB layer. The procedure follows the same steps as the original disentanglement experiment, but after swapping the unsupervised vectors s, we project the reconstructed latent w′  back into the VHCB latent space. If c and s are disentangled, both should be recovered with high accuracy after the swap.
>
> | Pseudo-label clf. | Concept set | Base Acc. c (no swap) | Acc. c (after swap) | Base  Acc. s (no swap) | Acc. s (after swap) |
> |--|--|--|--|--|--|
> | ResNet18 | all | 0.985 | 0.982 | 0.818 | 0.817 |
> | ResNet18 | balanced | 0.998 | 0.997 | 0.800 | 0.800 |
> | ResNet18 | low corr. | 0.997 | 0.997 | 0.805 | 0.805 |
>
> The ‘base accuracy’ reports the recovery of c and s from reconstructed latents obtained without swapping, where concept and side-channel vectors were generated jointly, serving as a baseline. The results show that swapping the concept and side-channel vectors between two samples does not affect recovery accuracy.
>
> - **DCI experiment**. Following the reviewer’s suggestion, we computed the DCI metric using the implementation provided at https://github.com/ema-marconato/glancenet, which is based on a Logistic Regression classifier. The metric was calculated using the concept vector c, the side-channel vector s, and the concatenation of both [c,s].
>
> | Pseudo-label clf. | Concept set | DCI c | DCI s | DCI [c,s] |
> |--|--|--|--|--|
> | ResNet18 | all | 0.163 | 0.095 | 0.161 |
> | ResNet18 | balanced | 0.207 | 0.043 | 0.201 |
> | ResNet18 | low corr. | 0.222 | 0.115 | 0.213 |
>
>   We observe two things: the DCI for s is significantly lower, indicating that concepts are not directly encoded in the unsupervised latent. Additionally, the DCI for c and for [c,s] is comparable, suggesting that s does not add information for the classifier, which continues to rely primarily on the features from c with the same entropy in the weights.

---

> ### Author Response · Authors · 2025-11-21
>
> **REFERENCES**
>
> María Martínez-García, Grace Villacrés, David Mitchell, and Pablo M. Olmos. Improved Variational
> Inference in discrete VAEs using Error Correcting Codes. In Proceedings of the Forty-first Conference on Uncertainty in Artificial Intelligence, volume 286 of Proceedings of Machine LearningResearch, pp. 2973–3012. PMLR, 21–25 Jul 2025.
>
> Aya Abdelsalam Ismail, Julius Adebayo, Hector Corrada Bravo, Stephen Ra, and Kyunghyun Cho. Concept Bottleneck Generative Models. In International Conference on Learning Representations, 2024.
>
> Akshay Kulkarni, Ge Yan, Chung-En Sun, Tuomas Oikarinen, and Tsui-Wei Weng. Interpretable
> Generative Models through Post-hoc Concept Bottlenecks. In IEEE/CVF Conference on Computer Vision and Pattern Recognition, 2025.
>
> Andrei Margeloiu, Matthew Ashman, Umang Bhatt, Yanzhi Chen, Mateja Jamnik, and Adrian
> Weller. Do Concept Bottleneck Models Learn as Intended? arXiv preprint arXiv:2105.04289,
> 2021.
>
> Marton Havasi, Sonali Parbhoo, and Finale Doshi-Velez. Addressing Leakage in Concept Bottleneck
> Models. Advances in Neural Information Processing Systems, 35:23386–23397, 2022.
>
> Joshua Lockhart, Nicolas Marchesotti, Daniele Magazzeni, and Manuela Veloso. Towards learn-
> ing to explain with concept bottleneck models: mitigating information leakage. arXiv preprint
> arXiv:2211.03656, 2022.
>
> Moritz Vandenhirtz, Sonia Laguna, Ričards Marcinkevičs, and Julia Vogt. Stochastic Concept Bottleneck Models. Advances in Neural Information Processing Systems, 37:51787–51810, 2024.
>
> Ao Sun, Yuanyuan Yuan, Pingchuan Ma, and Shuai Wang. Eliminating Information Leakage in
> Hard Concept Bottleneck Models with Supervised, Hierarchical Concept Learning. arXiv preprint
> arXiv:2402.05945, 2024.

---

> > ### Author Response · Authors · 2025-11-21
> >
> > *We would like to thank the reviewer for their positive and constructive feedback. We have carefully considered each comment, provided detailed responses above, and updated the paper accordingly. The submission has been revised to incorporate all of the reviewers’ suggestions. If the reviewer finds that their concerns have been adequately addressed, we respectfully invite them to update their score.*

---

### Official Review · Reviewer_AUUQ · 2025-11-01

**Soundness:** 3
**Presentation:** 3
**Contribution:** 3
**Rating:** 8
**Confidence:** 3

**Summary:**

This paper proposes the Variational Hard Concept Bottleneck (VHCB) layer that uses binary latent variables protected by error-correcting codes to map internal representations to hard human-interpretable concepts within pretrained generative models. Also, the authors introduce a systematic evaluation framework for assessing the steerability of CBGMs across various tasks, and demonstrate that VHCB consistently enhances steerability empirically. In general, the paper is clearly written, the metrics are sensible, and the authors also points out the impact of correlations and biases in training data, which is helpful for future work.

**Strengths:**

1. The authors introduce the limitations of existing approaches (i.e. relying on soft concept representations and susceptible to concept leakage), and the proposed VHCB layer is thus well motivated and is a clear conceptual advance.
2. The systematic evaluation framework for CBGMs proposed by this paper is thorough and comprehensive, which can be useful for many purposes.
3. The paper presents compelling empirical results that VHCB yields large gains in accuracy, and also transparently analyzes failures due to dataset correlations/bias.

**Weaknesses:**

1. The main comparison in this paper is VHCB vs. CB-AE. It would be helpful to compare the proposed method to several more baselines.
2. The authors use independent concept classifiers to obtain labels, which could introduce label bias that propagates into training and evaluation.

**Questions:**

1. Can you report a simple leakage diagnostic to empirically support the claim that hard concepts + compact s reduce leakage?
2. What would happen if you insert VHCB at multiple U-Net locations on DDPMs?

---

> ### Author Response · Authors · 2025-11-21
>
> **Additional baselines**
>
> To the best of our knowledge, only two prior works have addressed CBGMs, namely Ismail et al. (2024) and Kulkarni et al. (2025). Unfortunately, we were unable to successfully train models or reproduce the results of Ismail et al. (2024) using the official implementation provided by the authors, and thus could not include experimental results with this baseline. In their work, Kulkarni et al. (2025) reported encountering the same issue and therefore included the results from the original paper directly as a baseline: *“CBGM numbers are from their paper (1k samples) since their results are not reproducible using their released code. […] We could not compare with CBGM since they do not evaluate concept accuracy, and since we could not reproduce their results”.* However, this approach is not applicable in our case, as our experimental evaluation is more extensive and follows a different setup. We would greatly appreciate it if the reviewer could point us to any additional methods on CBGMs that could serve as suitable baselines.
>
> **Use of pretrained classifiers**
>
> We use pretrained classifiers (zero-shot or supervised) in a post-hoc framework, training the VHCB on images generated by the base generative model, which lacks concept annotations. This offers two main advantages: (i) flexibility in choosing the backbone architecture and (ii) reduced data requirements, as the original training set is not needed. We agree with the reviewer that strong biases in these classifiers could influence the model’s behavior, just as dataset biases affect the generative model itself. Alternatively, one could leverage the annotated dataset for training but this would restrict the choice of base architecture, since it requires access to triplets $(\boldsymbol{x}_i,\boldsymbol{w}_i,\boldsymbol{y}_i)$ during training. Since the VHCB maps generative embeddings w to concepts, it depends on latent embedding–concept pairs $(\boldsymbol{w}_i, \boldsymbol{y}_i)$ for supervision. For instance, a standard GAN cannot be directly applied in this setting, as its generator produces random images during training that cannot be associated with dataset annotations.
>
> **Concept leakage**
>
> Concept leakage occurs when concept probabilities unintentionally encode task-related information, in this case, additional generative factors. This may thus degrade steerability because interventions on a concept no longer produce the intended effect. For this reason, the steerability metrics in our analysis can in turn be seen as a proxy to assess leakage. Prior work on CBMs shows that hard concepts mitigate leakage due to their limited expressivity (Margeloiu et al., 2021; Havasi et al., 2022; Lockhart et al., 2022; Vandenhirtz et al., 2024; Sun et al., 2024). Our steerability experiments support this: compared to a baseline using soft concepts, our method achieves higher success rates for concept interventions, as shown by both quantitative (Tables 3, 6, 7, 8, 12) and qualitative evidence (Figures 4, 8, 9, 12).
>
> The side channel s is introduced to handle concept incompleteness, capturing generation-relevant factors not covered by the predefined concept set. Thus, as shown in the side-channel regularization ablation reported in our response to Reviewer 5vRH, its main effect lies on the generation quality (i.e., the FID metric). Yet, specially when the side channel size is large compared to the number of concepts, it is important to regularize the side channel to ensure that the decoder does rely on the concept set for generating images, and thus to ensure steerability.

---

> > ### Author Response · Authors · 2025-11-21
> >
> > **VHCB at multiple U-Net locations**
> >
> > We have extended our method by injecting the VHCB output into all levels of the U-Net upsampling path. Concretely, we replicate the structure of the U-Net upsampling path starting from the VHCB output and, for each upsampling block, introduce a small set of convolutional layers to project the VHCB representation to the required dimensionality. An updated architecture diagram is provided in the link below.
> >
> > The results obtained so far do not show clear visual improvements. Although we agree with the reviewer that this type of architecture is a promising direction, we also believe that it requires a more systematic architectural exploration and dedicated training, which we plan to pursue in future work. In the link below, we include three intervention examples for both VHCB-at-bottleneck and VHCB-at-all-levels. Specifically, we intervene on the following concepts: male → female, mouth slightly open (inactive → active), and smiling (inactive → active). As can be observed, the model incorporating VHCB at all upsampling levels achieves a visual performance very similar to the version in the original submission, where VHCB was applied only at the bottleneck. Once we finish running all training configurations, we will conduct the full set of evaluations and update the supplementary material with the complete metrics, together with the newly generated and intervened images, to provide a thorough comparison of the two designs.
> >
> > *Link to VHCB-at-all-levels and intervention figures:* https://postimg.cc/gallery/54cPNVg
> >
> > **REFERENCES**
> >
> > Aya Abdelsalam Ismail, Julius Adebayo, Hector Corrada Bravo, Stephen Ra, and Kyunghyun Cho. Concept Bottleneck Generative Models. In International Conference on Learning Representations, 2024.
> >
> > Akshay Kulkarni, Ge Yan, Chung-En Sun, Tuomas Oikarinen, and Tsui-Wei Weng. Interpretable
> > Generative Models through Post-hoc Concept Bottlenecks. In IEEE/CVF Conference on Computer Vision and Pattern Recognition, 2025.
> >
> > Andrei Margeloiu, Matthew Ashman, Umang Bhatt, Yanzhi Chen, Mateja Jamnik, and Adrian
> > Weller. Do Concept Bottleneck Models Learn as Intended? arXiv preprint arXiv:2105.04289,
> > 2021.
> >
> > Marton Havasi, Sonali Parbhoo, and Finale Doshi-Velez. Addressing Leakage in Concept Bottleneck
> > Models. Advances in Neural Information Processing Systems, 35:23386–23397, 2022.
> >
> > Joshua Lockhart, Nicolas Marchesotti, Daniele Magazzeni, and Manuela Veloso. Towards learn-
> > ing to explain with concept bottleneck models: mitigating information leakage. arXiv preprint
> > arXiv:2211.03656, 2022.
> >
> > Moritz Vandenhirtz, Sonia Laguna, Ričards Marcinkevičs, and Julia Vogt. Stochastic Concept Bottleneck Models. Advances in Neural Information Processing Systems, 37:51787–51810, 2024.
> >
> > Ao Sun, Yuanyuan Yuan, Pingchuan Ma, and Shuai Wang. Eliminating Information Leakage in
> > Hard Concept Bottleneck Models with Supervised, Hierarchical Concept Learning. arXiv preprint
> > arXiv:2402.05945, 2024.

---

> > > ### Author Response · Authors · 2025-11-21
> > >
> > > *We would like to thank the reviewer for their positive and constructive feedback. We have carefully considered each comment, provided detailed responses above, and updated the paper accordingly. The submission has been revised to incorporate all of the reviewers’ suggestions. If the reviewer finds that their concerns have been adequately addressed, we respectfully invite them to update their score.*

---

> > > ### Author Response · Authors · 2025-11-28
> > > **VHCB at multiple U-Net locations (CONTINUATION)**
> > >
> > > We trained the VHCB-at-all-levels model for different sets of concepts and summarized the main quantitative results in the table below. While the preliminary qualitative results in our previous response suggested that VHCB-at-all-levels achieved steerability similar to the VHCB-at-bottleneck, the aggregated quantitative metrics across concepts indicate that steerability generally improves when injecting the VHCB output at all levels of the U-Net upsampling path. We thank the reviewer for raising this point; based on these results, we confirm that this type of architecture is a promising direction for future research. However, fully leveraging this idea in score-based diffusion models requires a principled design of how the VHCB representation interacts with the multi-scale score network, including (i) how the control signal should be injected at different noise levels, (ii) how it should modulate the score during reverse-time SDE integration, and (iii) how to regularize cross-scale interactions to avoid inconsistencies across denoising stages. We consider these aspects essential for stabilizing training and achieving reliable control, and we plan to investigate them in future work.
> > >
> > >
> > > |Model | Pseudo Label Clf. | Concept Set | Concept Acc. | Acc intervention (i to a) | Acc intervention (a to i) |
> > > |----|----|----|----|----|----|
> > > |VHCB-at-bottleneck | ResNet18 | all | 0.733 | 0.131 | 0.546 |
> > > |VHCB-at-all-levels | ResNet18 | all | 0.753 | 0.150 | 0.568 |
> > > |----|----|----|----|----|----|
> > > |VHCB-at-bottleneck | ResNet18 | balanced | 0.546 | 0.406 | 0.250 |
> > > |VHCB-at-all-levels | ResNet18 | balanced | 0.567 | 0.562 | 0.313 |
> > > |---|---|---|---|---|---|
> > > | VHCB-at-bottleneck | ResNet18 | low corr | 0.621| 0.187 | 0.571 |
> > > | VHCB-at-all-levels | ResNet18 | low corr | 0.563 | 0.531 | 0.464 |

---

### Author Response · Authors · 2025-11-28

Dear Reviewers,

We believe we have carefully addressed all the concerns raised. If anything remains unclear or needs further clarification, we would be happy to provide more details, as we have time to do so. We look forward to your feedback, and if you think that our responses resolve your concerns, we respectfully invite you to update your score accordingly.

---

### Author Response · Authors · 2025-12-02

Dear AC,

Although the reviewers were unable to respond before the rebuttal period was unexpectedly cut short (even though we submitted our responses one week before, following the recommendations) we believe we have carefully addressed all concerns raised. Overall, the feedback on our work was positive (e.g., “the proposed VHCB layer is well motivated and a clear conceptual advance”, “the systematic evaluation framework for CBGMs proposed by this paper is thorough and comprehensive”). However, several clarifications and additional experiments were necessary to further support our claims. Below, we highlight the main points of discussion and how they were addressed during the rebuttal.

- **Mitigation of concept leakage.** We found necessary to clarify that leakage prevention primarily arises from the use of hard concepts (Margeloiu et al., 2021; Havasi et al., 2022; Lockhart et al., 2022; Vandenhirtz et al., 2024; Sun et al., 2024), as this was not entirely clear to the reviewers. Concept leakage occurs when concept probabilities unintentionally encode task-related information, in this case additional generative factors, which can degrade steerability because interventions on a concept no longer produce the intended effect. To assess leakage, we use steerability metrics as a proxy. Our qualitative and quantitative results confirm that using hard concepts improves the success rates of concept interventions compared to a baseline with soft concepts.

- **Disentanglement between concept vector and side channel (unsupervised latent).** Reviewer 5vRH requested additional evidence that the side channel captures residual information for generation rather than concept information. First, we obtained qualitative results, showing that concepts in the generated images are preserved when modifying the side channel. We also obtained additional quantitative results. As requested by the reviewer, we report the DCI scores for the latent representations, which indicate that concepts are not encoded in the unsupervised latent. To address concerns that our initial results might overestimate disentanglement due to the sensitivity of the pretrained classifiers, we repeated the experiment using the VHCB encoder (feedback loop experiment). By projecting the reconstructed latent w′ back into the VHCB latent space after modifying the side channel, we verified that both the concept vector and the side channel are accurately recovered after swapping, confirming effective disentanglement.

- **Effect of the regularization of the side channel (unsupervised latent).** Reviewer 5vRH requested an ablation study to investigate the impact of the regularization strength of the side channel (unsupervised latent) on overall model performance and steerability. We conducted experiments varying the weight of the unsupervised KL loss across different side-channel sizes and concept sets. The results indicate that this parameter has minimal effect on the reported metrics when using large concept sets. For smaller concept sets, both steerability and FID slightly degrade as side-channel regularization increases, since higher regularization constrains the model’s expressivity, reducing both generation quality and steerability. These findings are further supported by the experiment requested by Reviewer Te4N, where the side channel was entirely removed, resulting in a similar effect and a notable drop in generation quality.

- **Ablation on the concept loss.** Reviewer 5vRH requested further justification for our choice of symmetric KL divergence as the concept loss. To validate this choice empirically, we conducted an ablation study comparing models trained with BCE, KL, and symmetric KL as the concept loss. The results support the intuition presented in our submitted paper: models trained with symmetric KL achieve the best overall performance, demonstrating strong steerability in both directions (activation and deactivation of concepts).

- **VHCB at multiple U-Net locations.** Reviewer AUUQ suggested inserting the VHCB module at multiple levels of the U-Net architecture within the DDPM model. In response, we extended our method by injecting the VHCB output into all levels of the U-Net upsampling path (VHCB-at-all-levels) and trained this variant on different concept sets, providing both qualitative and quantitative results. While the initial qualitative results suggested that VHCB-at-all-levels performed similarly to VHCB-at-bottleneck, the aggregated quantitative metrics across concepts show that steerability generally improves when the VHCB output is injected at all levels of the upsampling path. We thank the reviewer for this suggestion; based on these results, we consider this architectural direction highly promising for future research.

---

> ### Author Response · Authors · 2025-12-02
>
> **References**
>
> Andrei Margeloiu, Matthew Ashman, Umang Bhatt, Yanzhi Chen, Mateja Jamnik, and Adrian Weller. Do Concept Bottleneck Models Learn as Intended? arXiv preprint arXiv:2105.04289, 2021.
>
> Marton Havasi, Sonali Parbhoo, and Finale Doshi-Velez. Addressing Leakage in Concept Bottleneck Models. Advances in Neural Information Processing Systems, 35:23386–23397, 2022.
>
> Joshua Lockhart, Nicolas Marchesotti, Daniele Magazzeni, and Manuela Veloso. Towards learning to explain with concept bottleneck models: mitigating information leakage. arXiv preprint arXiv:2211.03656, 2022.
>
> Moritz Vandenhirtz, Sonia Laguna, Ričards Marcinkevičs, and Julia Vogt. Stochastic Concept Bottleneck Models. Advances in Neural Information Processing Systems, 37:51787–51810, 2024.
>
> Ao Sun, Yuanyuan Yuan, Pingchuan Ma, and Shuai Wang. Eliminating Information Leakage in Hard Concept Bottleneck Models with Supervised, Hierarchical Concept Learning. arXiv preprint arXiv:2402.05945, 2024.

---

### Meta-Review · Area_Chair_zmtJ · 2025-12-28

**Summary:**

This submission introduces the Variational Hard Concept Bottleneck (VHCB), a probabilistic hard (binary) concept bottleneck layer for concept bottleneck generative models, motivated by the limitations of deterministic/soft concept bottlenecks for steerability and their susceptibility to concept leakage. The paper’s key contributions are (i) a probabilistic hard-concept formulation (implemented via a coded discrete VAE) that enables both concept interventions and direct generation from specified concept configurations, and (ii) a systematic steerability evaluation framework across multiple tasks (e.g., activation/deactivation, Hamming-distance-based interventions). Across the reviews, there is broad agreement that the problem is important and under-explored, the paper is clearly written, and the empirical evaluation is unusually thorough for this topic. In the rebuttal, the authors added targeted ablations and disentanglement diagnostics that directly address the main technical concerns (concept loss choice, side-channel regularization strength, and concept/side-channel disentanglement), and they also explored a stronger architectural variant that injects control at multiple U-Net locations. Based on the full record, I recommend Accept (poster).

**Reviewer Concerns:**

The primary concerns raised were: limited baseline coverage beyond CB-AE/closest prior CBGMs; whether the claimed leakage reduction is attributable to hard concepts versus side-channel regularization; whether the unsupervised side channel may itself carry concept information (and how to quantify independence/disentanglement beyond classifier-based checks); why symmetric KL is the appropriate concept loss; what role error-correcting codes (ECC) play with respect to leakage/steerability; and inconsistent FID comparisons and the notably worse FID when sampling random (independently sampled) concepts, which could undermine the “direct generation from concepts” contribution. The rebuttal substantially strengthened the empirical case by adding: an ablation comparing BCE vs (forward) KL vs symmetric KL for concept loss; an ablation over the unsupervised KL weight across side-channel sizes and concept sets; additional disentanglement evidence including qualitative side-channel swap results, a “feedback loop” recovery experiment using the VHCB encoder/decoder, and DCI-based measurements; and a side-channel removal experiment showing the expected quality/expressivity trade-off, especially for reduced concept sets. The authors also clarified that leakage mitigation is primarily due to hard concepts (with steerability metrics used as a practical proxy), while the side channel is intended to handle concept incompleteness and is regularized as a capacity-control mechanism. The remaining open point is largely about framing: random independent concept sampling can generate out-of-distribution concept combinations (inflating FID), so while this is not necessarily a flaw of the method per se, it suggests that practical “concept-to-image” sampling may benefit from modeling realistic joint concept distributions or constraining sampling; this limitation is now clearly articulated and does not outweigh the contributions for a poster acceptance.

**Reviewer Scores:**

AUUQ’s main requests (multi-location U-Net injection and additional leakage support) were addressed, and given the already strong initial stance, AUUQ would most plausibly remain at 8. Reviewer 5vRH’s key technical requests (concept-loss justification, KL-weight ablation, and more appropriate disentanglement/independence evidence) were addressed with concrete new experiments, making a modest upward revision plausible (most likely 6→7). Reviewer Te4N’s concerns about ECC’s role, the side channel, and the FID comparisons were directly discussed and partly validated via ablations; however, some skepticism may remain regarding random concept sampling and its implications, so a small upward revision is plausible (most likely 4→5, with some chance of staying at 4). Considering the strength of the contribution and the rebuttal’s substantive additions, the overall balance supports Accept (poster).

---

### Decision · Program_Chairs · 2026-01-26

Accept (Poster)